# Primase promotes the competition between transcription and replication on the same template strand resulting in DNA damage

Weifeng Zhang ®[1,2], Zhuo Yang[1,2], Wenjie Wang[1,2] & Qianwen Sun ®[1,2] ✉

Transcription-replication conflicts (TRCs), especially Head-On TRCs (HO-TRCs) can introduce R-loops and DNA damage, however, the underlying mechanisms are still largely unclear. We previously identified a chloroplast-localized RNase H1 protein AtRNH1C that can remove R-loops and relax HO-TRCs for genome integrity. Through the mutagenesis screen, we identify a mutation in chloroplast-localized primase ATH that weakens the binding affinity of DNA template and reduces the activities of RNA primer synthesis and delivery. This slows down DNA replication, and reduces competition of transcription-replication, thus rescuing the developmental defects of *atrnh1c*. Strand-specific DNA damage sequencing reveals that HO-TRCs cause DNA damage at the end of the transcription unit in the lagging strand and overexpression of ATH can boost HO-TRCs and exacerbates DNA damage. Furthermore, mutation of plastid DNA polymerase Pol1A can similarly rescue the defects in *atrnh1c* mutants. Taken together these results illustrate a potentially conserved mechanism among organisms, of which the primase activity can promote the occurrence of transcription-replication conflicts leading to HO-TRCs and genome instability.

Organelles in eukaryotic cells are considered the results of early bacterial endosymbiotic events[1]. Chloroplasts, the apparatus for photosynthesis, retain a genome of about 100–200 kb in size that comprises two inverted repeats (IR) separated by a small single-copy (SSC) region and a large single-copy (LSC) region[2]. Chloroplasts utilize nucleus-encoded proteins to conduct replication synchronously. Replication normally initiates from the origin, where DNA helicase separates DNA double helix and creates the replication fork utilizing energy from ATP hydrolysis[3]. DNA primase recognizes single-stranded DNA (ssDNA) to synthesize RNA oligonucleotides used as primers to initiate DNA replication. DNA polymerase utilizes the RNA primers and adds nucleotides matched to the template strand in the 5′ to 3′ direction[4]. While the leading strand is replicated continuously to the 3′ end of the complementary strand, the lagging strand in the opposite direction can only synthesize Okazaki fragments discontinuously as the

replication fork moves forward, which processes require the synthesis of RNA primers by primase[5].

The replisome of the chloroplast genome shares an evolutionary origin with that in bacteriophage T7[6–8], which consists of 4 proteins: the coupled primase–helicase (gp4), the single-stranded DNA binding protein (gp2.5), the DNA polymerase (gp5) and its processivity factor *E. coli* thioredoxin (trx)[9,10]. T7 gp4 protein is central to the replication machinery, it is composed of a zinc-binding domain (ZBD) that recognizes ssDNA template, an RNA polymerization domain (RPD) that adds ribonucleotide, and a helicase domain that unwinds double-stranded DNA[11]. Most eukaryotes own a homolog of the T7 gp4 protein named Twinkle (T7 gp4-like protein with intramitochondrial nucleoid localization)[12]. In metazoan organisms, Twinkle lacks the cysteines critical to zinc coordination in the ZBD domain and residues required for RNA synthesis in the RPD domain[12]. Instead, RNA polymerase

[1]Center for Plant Biology, School of Life Sciences, Tsinghua University, 100084 Beijing, China. [2]Tsinghua-Peking Center for Life Sciences, 100084 Beijing, China. ✉e-mail: sunqianwen@mail.tsinghua.edu.cn

synthesis transcripts are used as primers for DNA replication in metazoan mitochondria[13].

The Arabidopsis nucleus-encoded Twinkle homolog (Arabidopsis Twinkle homolog, ATH) is a 709-residue protein that localizes in chloroplasts and mitochondria[6]. Previous studies demonstrated that ATH has both primase and helicase activities in vitro[6]. ATH synthesizes RNA primers from a 5′-(G/C)GGA-3′ template sequence, and RNA oligonucleotides synthesized by ATH can be efficiently used as primers for plant organellar DNA polymerases Pol1A and Pol1B[14]. ZBD is typically associated with DNA binding domain[15]. In bacteriophage primase, two CXXC elements are essential to metal coordination in the ZBD domain[16]. In plant primase, the first CXXC repeat is conserved, but the second CXXC repeat is substituted by a CXRXKC element[14]. Previous studies indicate that the H33 and K70 residues located before the second CXXC repeat in T7, and *Clostridium difficile* primase drive ssDNA template recognition, respectively[16,17], and computer modeling of the ZBD domain in ATH indicated that the side chains of R166, K168, and W162 are in similar orientations than H33 of T7 primase and K70 of *C. difficile* primase[14] suggesting that the R166, K168, and W162 residues may be pivotal in ssDNA template recognition.

The machinery of DNA replication and RNA transcription operate on the same genome template, and collisions occur when they approach. The orientation of gene transcription relative to the direction of the replication fork determines the pattern of transcription-replication conflicts (TRCs). Studies in *Bacillus subtilis* and human cells revealed that head-on transcription-replication conflicts (HO-TRCs) induce the formation of stable R-loops and block fork progression, representing a major source of genomic instability[18–20]. Similar phenomena also happen in plants. Our previous studies revealed that the chloroplast-localized RNase H1 protein AtRNH1C can form a complex with chloroplast-localized DNA Gyrases (AtGyrases) and resolve HO-TRCs and R-loops in the rDNA HO-TRCs regions, thus maintaining genome integrity. Mutation of AtRNH1C leads to the formation of aberrant R-loops at these regions, causing genome breaks in chloroplast and growth defects[20,21]. By a reverse genetic screening, we also identified a DNA:RNA helicase, RHON1, as an R-loop resolvase operating in parallel with AtRNH1C to restrict HO-TRC-triggered R-loops and maintain genome integrity in chloroplasts, and the HO-TRC-trigged R-loops can be restricted by controlling the transcriptional activity of plastid-encoded RNA polymerases[20,21].

To uncover mechanisms how organisms could coordinate the transcription and replication with R-loop formation and genome maintenance, we adopted a forward genetic screening strategy to identify suppressors of *atrnh1c*. Here, we report that the primase of chloroplast genome replication, ATH, is responsible for enhancing HO-TRCs thus leading to R-loop accumulation and genome instability in *atrnh1c*. A point mutation in the zinc-binding domain (ZBD) weakens the binding of template DNA, and decreases the abilities of primer synthesis and delivery, thus slowing down replication and relieving transcription-replication competition. Over-expression of ATH leads to aberrant R-loop accumulation, which can be attenuated by the over-expression of AtRNH1C synchronously. Strand-specific DNA damage sequencing revealed that transcription-replication competition could introduce single-strand DNA breaks near the end of the transcription units. Furthermore, mutation of DNA polymerase Pol1A also can rescue the defects in *atrnh1c* by a similar mechanism. As HO-TRCs are commonly present in the genomes of all species, our results demonstrated a likely general mechanism of relaxing strand-specific transcription-replication competition to maintain genome integrity.

## Results

### A point mutation in *ATH* rescues the developmental defects of *atrnh1c*

The chloroplast-localized ribonuclease AtRNH1C plays a key role in maintaining plastid genome stability in Arabidopsis, and *atrnh1c*

displays pale yellow leaves, which is caused by aberrant R-loop accumulation and genome degradation[20]. To investigate new mechanisms regulating HO-TRCs, we conducted an ethyl methanesulfonate (EMS) mutagenesis screen and identified a suppressor of *atrnh1c*, named *acs1* (*atrnh1c suppressor 1*). Compared with *atrnh1c*, *acs1* showed the recovery of growth defects and leaf color in all stages of plant development, and the quantum efficiency of photosystem II (Fv/Fm), chlorophyll contents, the number and morphological features of chloroplasts in cells, and plant fresh weight all recovered to normal states (Fig. 1a, b and Supplementary Fig. 1).

With SHORE-map analysis[22] using the phenotypic recovered plants, we mapped a G to A point mutation in the fourth exon of At1g30680 that was highly correlated with phenotypic variation (Supplementary Fig. 2a, see methods). At1g30680 encodes a DNA primase-helicase with dual localization potential (chloroplast and/or mitochondria, Fig. 1c), for synthesizing RNA primers during organellar DNA replication[6]. Immunoblot analysis showed that the expression levels of chloroplast proteins RPOB, PetA, and RbcL were significantly reduced in *atrnh1c* compared to Col-0, while there was no difference in the expression of mitochondrial protein IDH (Supplementary Fig. 2b). In line with the phenotype, the levels of chloroplast proteins in the *acs1* double mutant were remarkably recovered to the level of Col-0 (Supplementary Fig. 2b). By constructing complementary vectors and genetic transformation, we confirmed that the mutation in At1g30680 is responsible for the phenotypic recovery in *acs1* (Fig. 1a, b and Supplementary Fig. 1).

The ATH protein contains a ZBD domain, an RNA polymerization domain (RPD), and a helicase domain, and the point mutation in At1g30680 leads to an amino acid change (R166K) in the ZBD domain (Fig. 1c). By protein structure prediction of Alphafold, we found that R166 is located at the end of a highly confidently predicted β-sheet (Supplementary Fig. 2c). Previous studies indicated that R166 might be vital for the ssDNA template recognition of ATH[14], and multiple sequence alignment revealed that R166 is conserved across land plants (Fig. 1d). EMSA and ChIP-qPCR assays showed that the R166K mutation in ATH indeed caused a significant reduction in the binding capacity to template DNA (Fig. 1e, f). Further genetic approaches confirmed that the complementary sequence with the point mutation could not complement the phenotype of *acs1*, confirming that the R166K mutation in *ATH* determines the phenotypic recovery in *acs1* (Supplementary Fig. 2d, e).

To directly assess the effect of the R166K mutation on primase activity, we compared the RNA primer synthesis ability of wild-type ATH and ATH(R166K) in vitro, and the results showed that the R166K mutation decreases primer synthesis (Supplementary Fig. 2g). We also detected the ATH and ATH(R166K) mediated RNA-primed DNA synthesis by Pol1A and Pol1B, the DNA polymerases located in plant organelles. When the wild-type ATH was used as primase in the reactions, the amount of synthesized DNA gradually increased as the amount of ATH increased. However, the DNA synthesis was even hard to detect using ATH(R166K) as the primase with equimolar amounts (Supplementary Fig. 2h). Moreover, we compared the helicase activities of the wild-type protein ATH and the mutant protein ATH(R166K), and found that the R166K mutation could affect the helicase activity of ATH (Supplementary Fig. 2i). In conclusion, these results indicate that the R166K mutation in ATH reduces the primer synthesis and delivery abilities as a primase.

### ATH is essential for plant development

To further investigate the functions of ATH, we analyzed its subcellular localization. Using transgenic plants of ATH-GFP, we found that the ATH protein displays a punctate distribution in chloroplasts (Fig. 2a, b). Protoplast transformation or tobacco leaf infection assays also showed that ATH is distributed in a punctiform manner in chloroplasts (Supplementary Fig. 3a, b), and the R166K mutation does not change

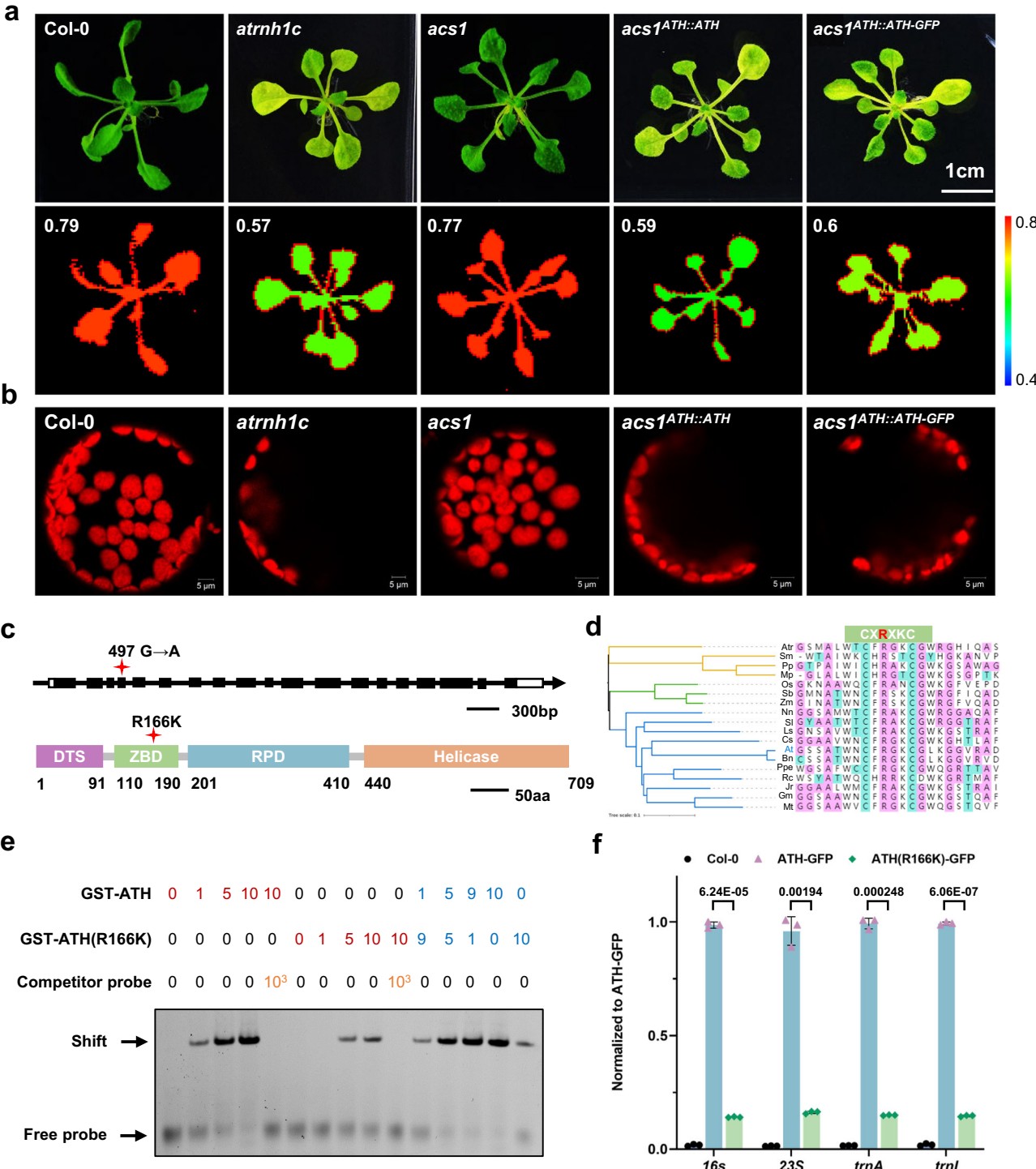

**Fig. 1 | A point mutation in *ATH* rescues the developmental defects of *atrnh1c* and weakens the binding of template DNA in replication. a** Photographs and chlorophyll fluorescence images of 21-day-old Col-0, *atrnh1c*, *acs1*, *acs1*^ATH::ATH, and *acs1*^ATH::ATH-GFP plants. Experiments were repeated three times with similar results. Chlorophyll Fv/Fm values are presented in the lower panel. Scale bar, 1 cm. **b** Comparison of chloroplasts in protoplasts from the leaves of 21-day-old Col-0, *atrnh1c*, *acs1*, *acs1*^ATH::ATH, and *acs1*^ATH::ATH-GFP plants. Chloroplasts are distinguished by chlorophyll autofluorescence (red). Scale bars, 5 μm. **c** Schematic representations of the *ATH* gene and its encoded protein. Exons are in black and 5′- and 3′-UTRs are in white. The positions of the point mutation and amino acid substitution are indicated. **d** Multiple sequence alignment between Twinkles in different plant species. The CXRXKC elements and R166 in ATH are shown. Proteins are designated with abbreviations as follows: Atr, *Amborella trichopoda*; Sm, *Selaginella*

*moellendorffii*; Pp, *Physcomitrium patens*; Mp, *Marchantia polymorpha*; Os, *Oryza sativa*; Sb, *Sorghum bicolor*; Zm, *Zea mays*; Nn, *Nelumbo nucifera*; Sl, *Solanum lycopersicum*; Ls, *Lactuca sativa*; Cs, *Cucumis sativus*; At, Arabidopsis thaliana; Bn, Brassica napus; Ppe, Prunus persica; Rc, *Rosa chinensis*; Jr, *Juglans regia*; Gm, *Glycine max*; Mt, *Medicago truncatula*. **e** The single-strand DNA binding activities of ATH and ATH(R166K) were evaluated by EMSA. Ten-hundred-fold unlabeled probes were used for competition. Equal amounts of the two proteins were used for the EMSA assays. Experiments were repeated three times with similar results. **f** ChIP-qPCR analysis showed the enrichment of ATH-GFP and ATH(R166K)-GFP at cp-rDNA regions. Col-0 was used as the negative control. Three biological replicates were performed. The graphs represent the mean ± SD. **P < 0.01; ***P < 0.001 by unpaired two-sided t test. Source data are provided as a Source Data file.

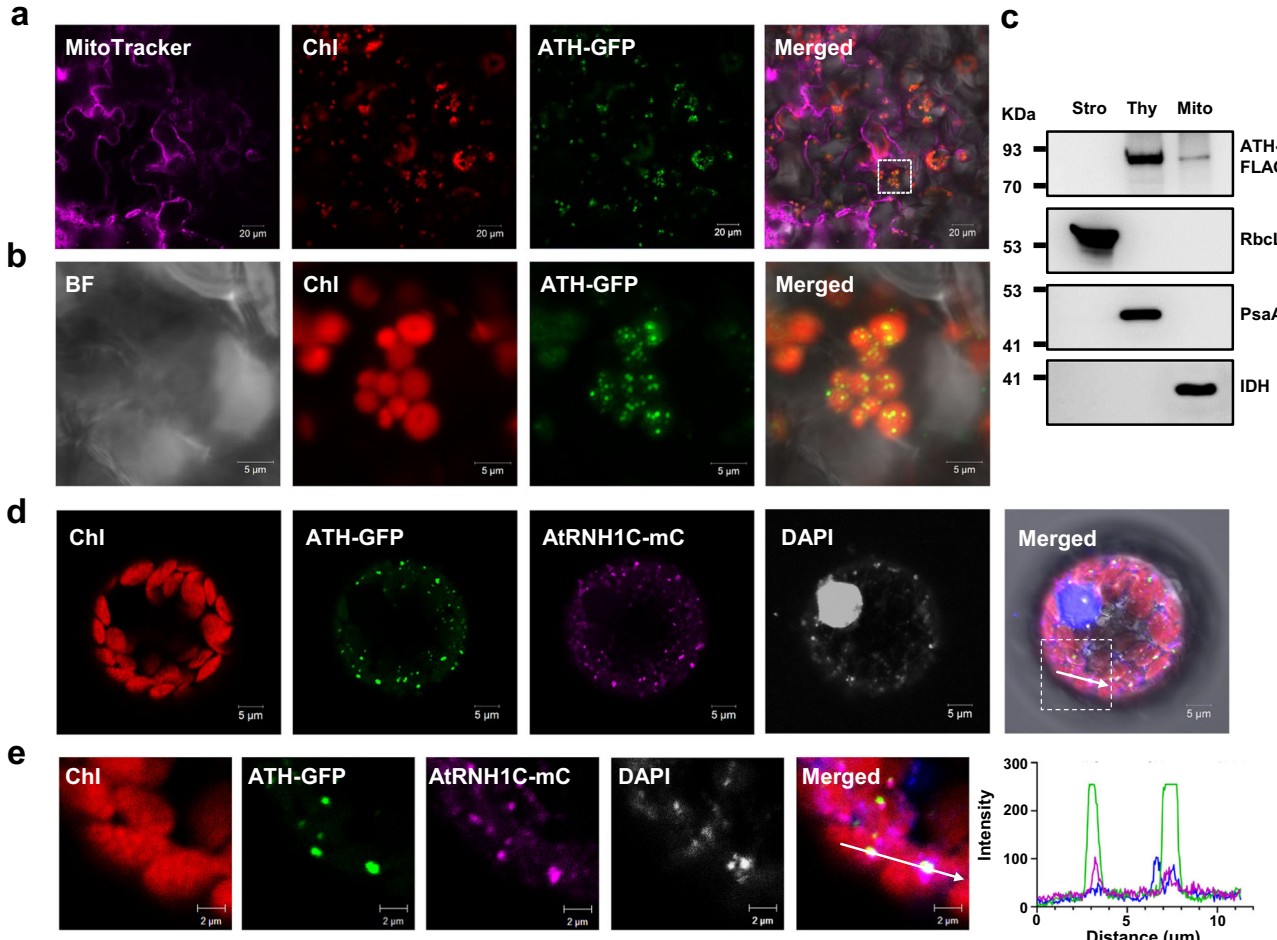

**Fig. 2 | ATH is co-localized with AtRNH1C in the thylakoids of chloroplasts.**
**a** Subcellular localization of GFP fused to the full-length ATH protein in the leaves of transgenic plants. MitoTracker, mitochondria marker (magenta); Chl, chlorophyll autofluorescence (red); ATH-GFP, ATH fused with GFP (green). The white box indicates the region magnified in (**a**). Scale bars, 20 μm. Experiments were repeated three times with similar results. **b** Magnified images of the boxed areas in (**a**). Scale bars, 5 μm. BF, bright field. **c** Immunoblots of protein fractions isolated from chloroplast stroma (Stro), thylakoids (Thy), and mitochondria (Mito) of ATH-FLAG transgenic plants. The ATH-FLAG protein was detected by the Anti-FLAG monoclonal antibody, and the polyclonal antibodies anti-RbcL, anti-PsaA, and anti-IDH1 were used to indicate stromal, thylakoid, and mitochondrial protein fractions, respectively. Experiments were repeated three times with similar results. **d** Co-localization of ATH-GFP and AtRNH1C-mC (mCherry) in Arabidopsis protoplasts. DAPI shows the nucleus (gray); scale bars, 5 μm. Experiments were repeated three times with similar results. **e** Magnified images of the boxed areas in (**d**). Scale bars, 2 μm. The right image is a line scanned at the position depicted by the white line. Source data are provided as a Source Data file.

the distribution pattern of ATH (Fig. S3A). The ATH protein could also be detected in mitochondria (Figs. S3A and 3C), which is consistent with previous results[23]. Western blot assays revealed that the ATH protein is mainly localized in the thylakoids of chloroplasts and mitochondria (Fig. 2c). The RNase H1 protein AtRNH1C was also found to be enriched within the thylakoids, where the nucleoids are located[20]. By protoplast co-transformation, we observed a high co-localization distribution of ATH and AtRNH1C in chloroplasts (Fig. 2d, e). The co-localization of ATH with AtRNH1C on the chloroplast thylakoids implies that there might be some functional correlations between them.

Considering the bidirectional localization property of ATH, to exclude the effect of mitochondrial localization of ATH on the phenotype, we replaced the ATH signal peptide with the reported signal peptides of AtRNH1C and RecA1 proteins, which were specifically localized in chloroplasts, and the genetic complementation experiments showed that ATH specifically localized to chloroplasts could also effectively recover the phenotype of *acs1* (Supplementary Fig. 3d).

We further observed the tissue distribution of ATH using transgenic complementary plants with GUS (β-glucuronidase) fusion. The ATH protein was mainly distributed in young meristematic tissue regions such as SAM (shoot apical meristem), young leaves, root tips, inflorescences, and young siliques (Supplementary Fig. 4). These tissues are the places where active DNA replication is present, and thus we speculate that this expression property of ATH is highly correlated with its function in DNA replication.

The mutant with the R166K mutation in the Col-0 background (named *ath-1*) exhibited a growth retardation phenotype in the early growth stage compared to Col-0 (Supplementary Fig. 5a), but with growth, the difference between the two is no longer obvious (Supplementary Fig. 5b). A previous study showed that a T-DNA insertion mutant (SALK_152246) of *ATH* gene displays a phenotype similar to the wild-type plants (Supplementary Fig. 5b)[8]. We examined the position of T-DNA insertion in the SALK_152246 line by sequencing, and found that the T-DNA insertion occurs in the 5′ UTR region of the *ATH* gene (Supplementary Fig. 5c, d). RT-qPCR assays also showed that the T-DNA insertion does not affect the *ATH* expression (Supplementary Fig. 5e). We then tried to create knockout mutants of ATH by the CRISPR-Csa9 system[24] to further verify its function. However, we were unable to obtain any *ath* homozygous knockout mutants, either in Col-0 or *atrnh1c* background. Instead, we obtained several lines of *ath*+/− heterozygotes (Supplementary Fig. 6). Therefore, the knockout

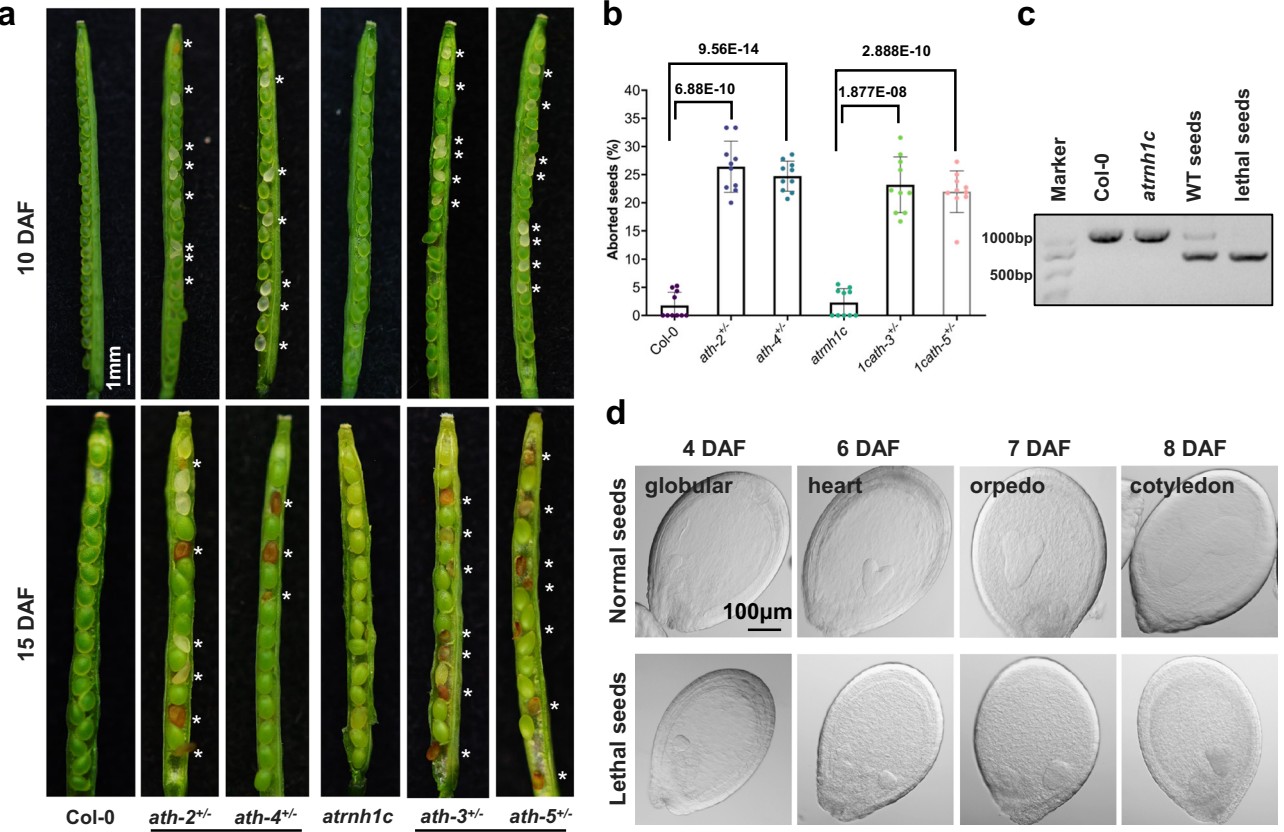

**Fig. 3 | Knockout of *ATH* leads to embryo lethality. a** Seed development in the siliques of wild-type and *ath*[+/−] plants at 10 days (upper panel) and 15 days (lower panel) after flowering (DAF). The asterisks (*) indicate the aborted seeds. Scale bar, 1 mm. **b** Percentage of aborted seeds in the siliques shown in (**a**). Data were calculated from 10 siliques for each genotype, indicated by individual dots. The graphs represent the mean ± SD. ****$P < 0.0001$ by unpaired two-sided *t* test. **c** Genotyping of wild-type and lethal seeds in the siliques shown in (**a**). Experiments were repeated three times with similar results. **d** Embryo development in wild-type and aborted seeds of the siliques of *ath*[+/−] plants. DAF, days after flowering. The wild-type embryos showed normal developmental processes: globular-shaped (4 DAF), heart-shaped (6 DAF), torpedo-shaped (7 DAF), and cotyledon-shaped (8 DAF). Scale bar, 100 μm. Experiments were repeated three times with similar results. Source data are provided as a Source Data file.

mutation of ATH might be embryonic lethal. We observed the seeds inside the siliques of *ath*+/− heterozygous deletion plants and found that some of the seeds were white, turning brown and crumbling as they developed (Fig. 3a). The percentage of these problematic seeds in the siliques was calculated to be approximately 25%, while the seeds in either Col-0 or *atrnh1c* were completely normal (Fig. 3b). By genotyping, we found that the lethal seeds were *ath* homozygous (Fig. 3c). Further observation found that the development of problematical seeds stayed at the globular stage (Fig. 3d). These results indicated that ATH is indispensable for plant development and that knockout of ATH leads to embryonic lethality.

We analyzed the phenotypes of the CRISPR/Cas9-generated *ath*[+/−] heterozygous and *1cath*[+/−] plants, and the *1cath-3*[+/−] plants displayed partially rescued yellowish phenotypes compared to the *atrnh1c* mutant, while the phenotype of *1cath-5*[+/−] plants recovers slightly (Supplementary Fig. 6f). The recovery could be because of the loss of ATH protein levels, as the heterozygous mutants carry only one copy of functional *ATH* gene (Supplementary Fig. 6g). Furthermore, by S9.6 slot-blot assay, we found a significant decrease of R-loops in the chloroplast of *1cath*[+/−] plants compared to that of *atrnh1c* (Supplementary Fig. 6h).

### *acs1* relieves HO-TRCs and restricts R-loop accumulation

To investigate the mechanism of phenotypic recovery caused by the ATH(R166K) mutation, we first examined the R-loop levels in *acs1* by

chloroplast S9.6 slot-blot, DRIP-qPCR, and immunostaining. It has been previously reported that S9.6 may not be optimal for immunostaining to detect RNA:DNA hybrid due to possible interference from dsRNA. However, binding affinity measurements in new studies showed S9.6 exhibits specificity for DNA-RNA hybrid over dsRNA[25,26], and our previous immuno-staining assays in chloroplast also showed dsRNA does not affect the R-loop recognitions specificity of S9.6 antibody[21,27]. The results all showed a significant decrease in R-loop levels in *acs1* chloroplasts compared to *atrnh1c*, while R-loop levels in complementary materials were comparable to those in *atrnh1c* (Fig. 4a–c). As R-loop accumulation triggers chloroplast genome instability in *atrnh1c*[20], we investigated whether the decreased level of R-loops in *acs1* could relieve genome degradation in the chloroplast. The PFGE (pulsed-field gel electrophoresis) results showed that consistent with previous results[20], the monomeric and oligomeric cpDNA (chloroplast DNA) molecules degraded dramatically in *atrnh1c* compared with Col-0, and these forms were significantly restored in *acs1*, with a remarkable decrease in degraded DNA molecules (Fig. 4d). TUNEL (terminal deoxynucleotidyl transferase dUTP nick end labeling) and neutral comet assays also confirmed the decrease in DNA damage in *acs1* compared to *atrnh1c* (Supplementary Fig. 7a, b).

We then analyzed the transcription and replication states in *acs1*. Bioanalyzer (Agilent 4200) results showed that the mature cp-rRNAs were recovered in *acs1* compared to *atrnh1c* (Fig. 4e). The expression of chloroplast rRNA transcription intermediates and mature rRNAs

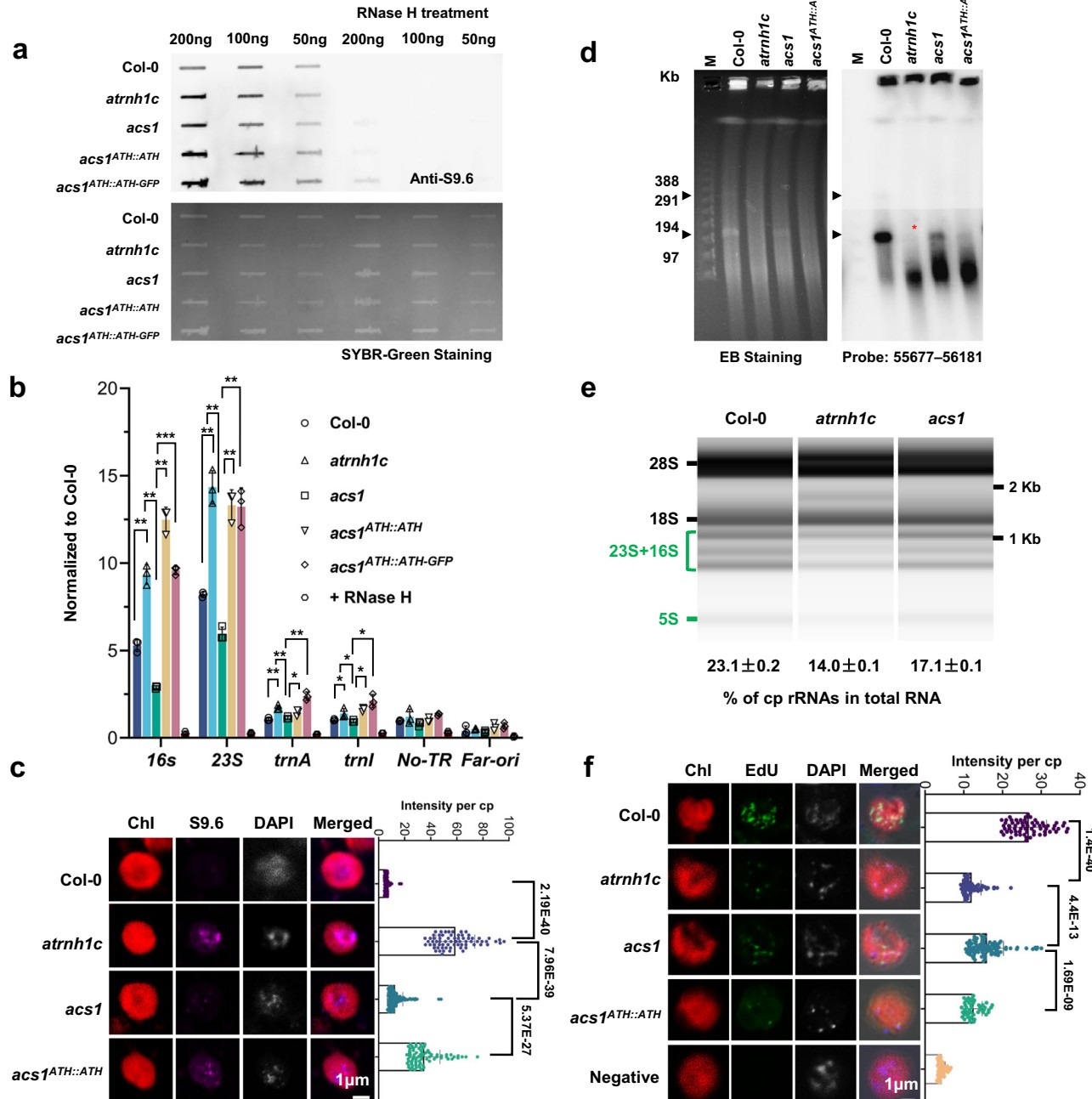

**Fig. 4 | *acs1* relieves HO-TRCs and restricts R-loop accumulation. a** Slot-blot assays of the overall chloroplast R-loop levels of 21-day-old Col-0, *atrnh1c, acs1, acs1^(ATH::ATH)* and *acs1^(ATH::ATH-GFP)* plants. RNase H-treated DNA was used as a negative control. R-loops were detected using the RNA:DNA hybrid antibody S9.6 (up panel). DNA loading was stained using SYBR Green (down panel). Experiments were repeated three times with similar results. **b** R-loop levels in the cp-rDNA head-on regions were detected by DRIP-qPCR. RNase H treated *atrnh1c* cpDNA sample was used as a negative control. Three biological replicates were performed. The graphs represent the mean ± SD. \**P* < 0.05; \*\**P* < 0.01; \*\*\**P* < 0.001 by unpaired two-sided *t* test. **c** R-loop signal (magenta) accumulation in chloroplasts was detected by S9.6 immunostaining. DAPI was used to observe nucleoids in chloroplasts. Representative images are shown on the left. *n* = 69. The graphs represent the mean ± SD. \*\*\*\**P* < 0.0001 by unpaired two-sided *t* test. Scale bar, 1 μm. **d** PFGE assay of cpDNA

from 21-day-old Col-0, *atrnh1c, acs1* and *acs1^(ATH::ATH)* plants. The left panel shows ethidium bromide staining, and the right panel shows blot hybridization of probe 55677–56181 (a 505-bp *rbcL* gene fragment). A Lambda Ladder (New England Biolabs; N0341) was used to indicate the molecular weight. Arrowheads indicate the structures of cpDNA monomers and dimers. The red asterisk shows the recovery of cpDNA monomers in *acs1*. Experiments were repeated three times with similar results. **e** Bioanalyzer (Agilent 4200) results showing total RNAs isolated from 21-day-old plant leaves. The calculation of the percentage of cp-rRNAs in total RNAs is shown at the bottom. Experiments were repeated three times with similar results. **f** EdU labeling was used to measure the intensity of DNA replication in chloroplasts. Representative images are shown on the left. Scale bar, 1 μm. *n* = 45. The graphs represent the mean ± SD. \*\*\**P* < 0.001; \*\*\*\**P* < 0.0001 by unpaired two-sided *t* test. Source data are provided as a Source Data file.

was also calculated by RT-qPCR, and the results showed that the level of rRNA transcripts in *acs1* chloroplasts was significantly elevated compared to that in *atrnh1c* (Supplementary Fig. 7c). We next applied EdU (5-ethynyl-2′-deoxyuridine) staining to assess the replication state

in the chloroplast[21]. As with the phenotypic changes, significant DNA replication stress alleviation was observed in *acs1* compared to *atrnh1c* (Fig. 4f). We then conducted a DAPI staining method to analyze DNA replication states by investigating patterns of nucleoids[28]. In line with

previous results, a large percentage of chloroplasts from *atrnh1c* owned large extended nucleoids (type III), whereas most nucleoids in *acs1* displayed scattered distribution (type I or II, Supplementary Fig. 7d), indicating that the chloroplast genome replication stress was mitigated in *acs1* comparing to *atrnh1c*. Furthermore, 2D gel electrophoresis showed that the replication intermediates of the cp-rDNA region were recovered in *acs1* compared with *atrnh1c* (Supplementary Fig. 7e). Taken together, these results imply that the R166K point mutation of ATH alleviates the excessive accumulation of R-loops in *atrnh1c*, thereby partially restoring the defects of transcription and replication, and maintaining genomic stability in the chloroplast.

## HO-TRCs cause single-strand DNA breakage at the end of transcription units

As HO-TRCs could result in R-loop accumulation and DNA damage, we then analyzed the DNA damage sites in the chloroplast genome by DEtail-seq (DNA End tailing and sequencing), a method we recently developed that can detect 3′ end damage sites with strand-specific information[29]. Along with this experiment, we also analyzed the distribution of the R-loop in the chloroplast genome by ssDRIP-seq[30], together with the binding profile of ATH by ChIP-seq. Compared with Col-0, the strand breaks in the *atrnh1c* chloroplast genome were significantly increased, while the damages in the *acs1* genome were decreased compared to those in *atrnh1c*, with only high signals present in the high transcription-replication collision regions, where R-loop levels were highly accumulated (Fig. 5a–c). Although the DNA breaks were recovered in the double mutant *acs1* compared to *atrnh1c*, they were still higher than those in Col-0 (Fig. 5a–c, Supplementary Fig. 8a–c). Of particular note, single-strand DNA breaks (SSBs) were dramatically enriched on the strand that templates transcription, especially at the end of transcription units, where high HO-TRCs happened (Fig. 5a–c, Supplementary Fig. 8a–c). This pattern of strand-specific SSBs could also be detected in wild-type Col-0 (Fig. 5a–c, Supplementary Fig. 8a–c), although the level was much lower than that in *acs1*. These results suggested that the primase ATH is mainly for lagging strand replication, and enhances HO-TRCs thus leading to R-loop accumulation and genome instability through HO-TRCs in high transcription regions of the chloroplast genome (Fig. 5d), and the R166K mutation in ATH slows down DNA replication, relieves HO-TRCs and rescues genome integrity in *atrnh1c*.

Another noteworthy phenomenon is that the overall distribution pattern of chloroplast DNA breaks in *atrnh1c* is similar to the pattern of the ATH binding profile (Fig. 5a). Previous studies indicated that RNase H is involved in the removal of RNA primers[31–33], and RNA primers were excessively accumulated in RNase H mutants[31]. We hypothesize that in *atrnh1c*, the inability to remove RNA primers from DNA templates leads to the over-accumulation of short RNA:DNA hybrids, which triggers extensive breaks in the chloroplast genome. To test this speculation, we examined the degradation ability of AtRNH1C for RNA primers in vitro, and the results showed that AtRNH1C was able to efficiently remove small fragments of RNA from the DNA template (Supplementary Fig. 8d). Thus, these results indicated that in chloroplasts, AtRNH1C may also function to scavenge RNA primers generated during DNA replication in addition to degrading transcriptionally generated R-loops (Fig. 5d), and the massive accumulation of RNA primers may be an important trigger for extensive breaks in the chloroplast genome of *atrnh1c* mutant.

## ATH antagonizes R-loop clearance machinery to strengthen HO-TRCs and boost DNA damage

To further prove the working model, we artificially overexpressed ATH in Col-0 and a*cs1*, and both resulted in the pale-yellow young leaves of the plants (Fig. 6a), with the number of chloroplasts in cells and the quantum efficiency of photosystem II (Fv/Fm) significantly reduced (Fig. 6b, d, e). The effects became more pronounced as the plants grew

and developed (Supplementary Fig. 9a). Consistent with the phenotype, in the ATH overexpression plants, the R-loops were over-accumulated in chloroplasts (Fig. 6c, f–h), and the stability of the chloroplast genome was significantly reduced (Fig. 6i, Supplementary Fig. 9b), companied by a much higher level of DNA damage (Fig. 6j, Supplementary Fig. 9c). In addition, overexpression of ATH also led to transcription and replication inhibition, showing a decrease in cp-rRNA transcription and DNA replication (Supplementary Fig. 9d–g).

S9.6 immunostaining of ATH-GFP overexpression plants showed that the GFP signals were highly co-localized with the S9.6 signals in the chloroplast nucleotides (Fig. 6k). Furthermore, we analyzed the co-localization of ATH with RNA:DNA hybrids by co-transforming ATH-GFP with inactive AtRNH1C(D222N)-mCherry, and results showed that the co-localization of the two proteins can be seen in the nucleoids (Supplementary Fig. 9h). Overexpression of AtRNH1C in ATH over-expressing plants alleviated the phenotype of yellowish leaves (Supplementary Fig. 10). These results showed that the growth defects caused by overexpression of ATH could be relieved by overexpressing AtRNH1C, further confirming an antagonistic function of primase ATH and R-loop clearance machinery that AtRNH1C involved.

Our previous study found that the R-loop helicase RHON1 is also involved in R-loop clearance to maintain chloroplast genome integrity, and the *atrnh1c/rhon1* double mutant (*1crhon1*) displays more severe phenotypic defects than *atrnh1c*[21]. To investigate whether the R166K mutation in ATH can also rescue the phenotype of *1crhon1*, we crossed the *rhon1* and *acs1* mutants to generate the *rhon1acs1* triple mutant (Supplementary Fig. 11a). Compared with *1crhon1*, the plant size, photosystem II efficiency, and the number of chloroplasts per cell of the *rhon1acs1* triple mutant were all partially restored (Supplementary Fig. 11a–d). S9.6 slot-blot and immunostaining showed a decrease in the R-loop level in the *rhon1acs1* triple mutant compared to *1crhon1* (Supplementary Fig. 11e, f), and the TUNEL assay indicated that the DNA damage in the *rhon1acs1* triple mutant was weaker than that of *1crhon1* (Supplementary Fig. 11g). The transcription and replication levels in the *rhon1acs1* triple mutant also recovered (Supplementary Fig. 11h, i). All these results indicated that the R166K mutation of ATH can also rescue the phenotype of the *1crhon1* double mutant by restricting HO-TRCs thus relieving R-loop accumulation and HO-TRCs. Naturally, the primase ATH and R-loop clearance machinery are antagonistic to each other to balance HO-TRCs and genome integrity.

## Mutation of Pol1A also can rescue the growth defects of *atrnh1c* by restricting HO-TRCs in chloroplasts

Replication of the chloroplast genome requires the involvement of two DNA polymerases, Pol1A and Pol1B[34]. Previous studies showed that Pol1B is not only involved in replication but also has functions of DNA repair[35], and the double mutant of *pol1b* and *atrnh1c* displayed more severe chloroplast genome degradation and growth defects than *atrnh1c*[36]. To further investigate the effect of DNA replication with R-loop accumulation and genome stability, we constructed a *pol1a* and *atrnh1c* double mutant (*1cpol1a*) (Supplementary Fig. 12a). Compared with *atrnh1c*, the growth defects, photosystem II efficiency, chlorophyll contents, and chloroplast number per cell in leaves of the *1cpol1a* double mutant were partially restored (Fig. 7a–e). DRIP-qPCR and S9.6 immunostaining also showed the R-loop level in the *1cpol1a* double mutant decreased compared to *atrnh1c* (Fig. 7f, g, and Supplementary Fig. 12b). In line with the phenotype, TUNEL assay, PFGE, and Detail-seq results showed that the DNA damage and genome degradation in *1cpol1a* also decreased than that of *atrnh1c* (Fig. 7h, i, and Supplementary Fig. 12c–e). These results further confirmed that weakening DNA replication could relieve HO-TRCs and maintain genome integrity.

## Discussion

Transcription and replication are the most essential events of living organisms to sustain life, and they rely on the same genome as the

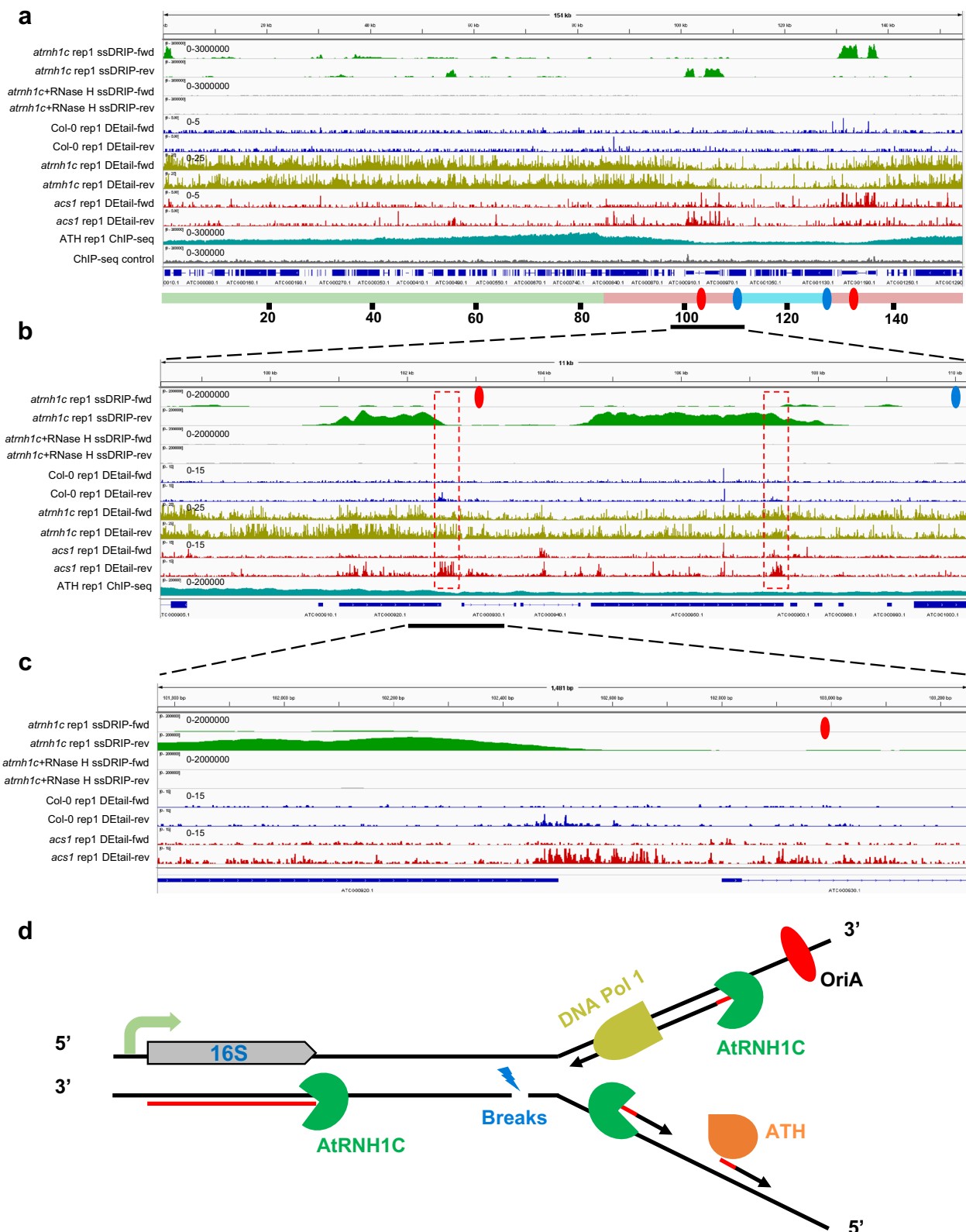

**Fig. 5 | HO-TRCs cause single-strand DNA breaks at the end of the transcription units. a** Snapshots of ssDRIP-seq, DEtail-seq, and ATH ChIP-seq in the whole genome of the chloroplast. The green, blue, and pink sticks show a large single-copy (LSC) region, a small single-copy (SSC) region, and two inverted repeats (IR) regions, respectively. The red and blue ellipses show OriA and OriB. One of two replicates is shown. **b** Snapshots of ssDRIP-seq, DEtail-seq, and ATH ChIP-seq in the rDNA region of the chloroplast indicated by the overline in (**a**). **c** Snapshots of ssDRIP-seq and DEtail-seq in the area indicated by the overline in (**b**). **d** Schematic representation of the HO-TRCs competition and DNA breaks in the rDNA region of the chloroplast genome.

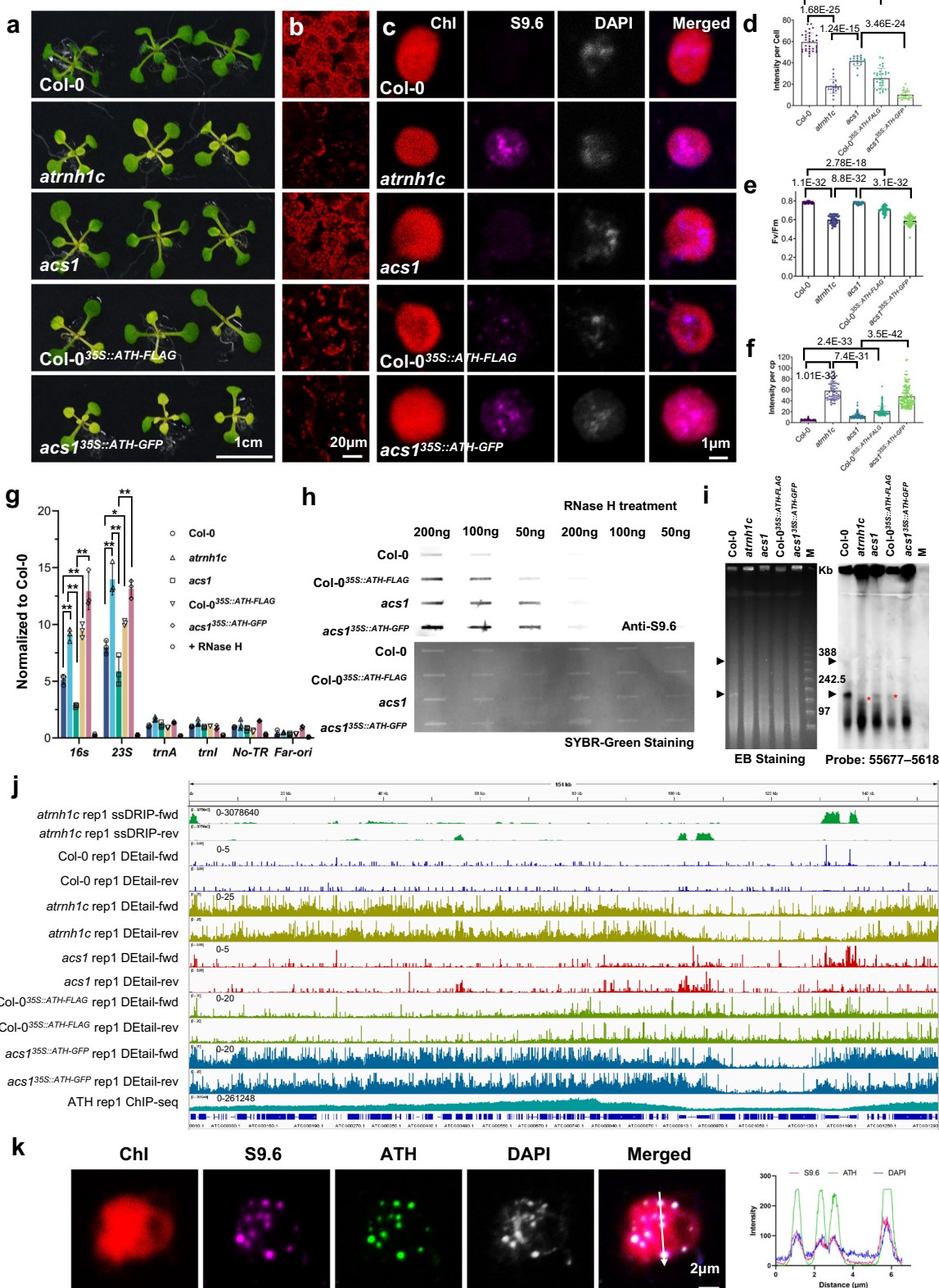

template. According to the coding strand of gene transcription relative to the movement of the replisome, the transcription and replication machinery either move head-on or codirectionally, which determines the pattern of transcription-replication conflicts. HO-TRCs induce R-loops and compromise replication and the expression of head-on genes, thus triggering DNA breaks and genome instability[37–41]. In bacteria, genomes have evolved to organize the majority of genes

expressed codirectionally with replication forks, thus avoiding head-on collisions[42]. However, this is not the case in semiautonomous chloroplasts.

Previous results found that the replication origins in the chloroplast genome are located inside the highly transcribed rDNA regions in the IR regions[43]. This localization feature creates natural HO-TRCs (Fig. 5d). As the rDNAs are the highest transcribed regions in the

**Fig. 6 | Overexpression of ATH boosts HO-TRCs and enhances DNA damage.**
**a** The phenotypes of 14-day-old Col-0, *atrnh1c*, *acs1*, Col-0*3SS::ATH-FLAG*, and *acs1*3SS::ATH-GFP plants. Scale bar, 1 cm. **b** Cytological observation of chloroplasts in the leaves of 21-day-old plants. Scale bar, 20 μm. Experiments were repeated three times with similar results. **c** R-loop signal (magenta) accumulation in chloroplasts was detected by S9.6 immunostaining. Scale bar, 1 μm. Experiments were repeated three times with similar results. **d.** The intensity of chlorophyll autofluorescence per leaf cell of different plants. *n* = 19. Statistical testing was performed by unpaired two-sided *t* test. **e** Chlorophyll Fv/Fm values of plants are indicated in (**a**). *n* = 40. Statistical testing was performed by unpaired two-sided *t* test. **f** R-loop signal intensity in chloroplasts detected by S9.6 immunostaining. *n* = 60. Statistical testing was performed by unpaired two-sided *t* test. **g** R-loop levels in the cp-rDNA head-on regions were detected by DRIP-qPCR. The graphs represent the mean ± SD. Statistical testing was performed using unpaired two-sided *t* test. *$P < 0.05$; **$P < 0.01$.

**h** Slot-blot assays of the overall chloroplast R-loop levels of 21-day-old plants. RNase H-treated DNA was used as the negative control. Experiments were repeated three times with similar results. **i** PFGE assay of cpDNA from 21-day-old plants. The left panel shows ethidium bromide staining, and the right panel shows blot hybridization of probe 55677–56181 (a 505-bp *rbcL* gene fragment). A Lambda Ladder (New England Biolabs; N0341) was used to indicate the molecular weight. Arrowheads indicate the structures of cpDNA monomers and dimers. The red asterisks show the reduction of cpDNA monomers in Col-0*3SS::ATH-FLAG* and *acs1*3SS::ATH-GFP. Experiments were repeated three times with similar results. **j** Snapshots of DEtail-seq in the whole chloroplast genome of 21-day-old plants. One of two replicates is shown. **k** Images of ATH and AtRNH1C co-localization by chloroplast immunofluorescence analysis. The right panels are line scans at the position depicted by the corresponding white lines. Scale bar, 2 μm. Experiments were repeated three times with similar results. Source data are provided as a Source Data file.

chloroplasts, and with two origins located inside, this could induce much higher risks of head-on transcription-replication conflicts. By investigating multiple pathways that restrict HO-TRCs-promoted R-loops, we previously found that AtRNH1C and RHON1 synergistically restrict R-loops and release transcription and replication, thus supervising chloroplast genome integrity and ensuring the normal development of plants[20,21]. Here, through a phenotypic suppressor screen of *atrnh1c*, we found that an R166K mutation in the ZBD domain of the chloroplast-localized primase ATH can rescue the growth defects in *atrnh1c*. Further investigation revealed that the R166K mutation decreases the RNA primer synthesis and delivery activities, slowing down the replication machinery and relieving HO-TRCs, thus reducing R-loops and DNA damage in the chloroplast genome. These results reveal that the chloroplast primase ATH plays a vital role in R-loop coordination and genome integrity maintenance. Furthermore, mutation of Pol1A, one of the two DNA polymerases in plant organelle, can also lead to similar effects in rescuing the defects in *atrnh1c* mutant. These results indicate that the HO-TRCs can be mitigated by reducing the DNA replication speed, which may be a common mechanism across species.

By DNA breaks sequencing in the chloroplast, we found the DNA breaks were enriched in the lagging strand, especially at the end of transcription units in Col-0 and *acs1*. This pattern of strand-specific DNA breaks suggested that the primase ATH, which is mainly for lagging strand replication, enhances R-loop accumulation and genome instability at the HO-TRCs regions. We also found the distribution pattern of DNA breaks in *atrnh1c* is similar to the binding pattern of ATH protein (Fig.5a). Previous studies revealed that RNA primers were excessively accumulated in RNase H mutants[31]. By testing the ability of AtRNH1C to digesting RNA primers, we found that AtRNH1C was able to efficiently remove small fragments of RNA from DNA template (Supplementary Fig. S8d). Thus, we hypothesize that in chloroplasts, AtRNH1C can also degrade RNA primers from DNA templates during replication. In *atrnh1c*, the inability to remove RNA primers from DNA templates leads to the over-accumulation of RNA:DNA hybrids, which triggers extensive breaks in the chloroplast genome. Indeed, over-expression of ATH in wild type and *acs1* also leads to more DNA breaks in the chloroplast genome and plant growth defects, which can be rescued by over-expression of AtRNH1C synchronously (Fig.6, Supplementary Fig. 9, and Supplementary Fig. 10). These genetic and sequencing results further confirmed the hypothesis that AtRNH1C could remove the RNA primers during replication, especially in the RNA primers enriched lagging strand.

Strand-specific mutations occur in the process of DNA replication and transcription. Discontinuous synthesis of the lagging strand during replication produces a series of Okazaki fragments, the 5′ ends of which have increased levels of nucleotide substitution[44]. Additionally, longer exposure as ssDNA may cause the lagging strand to be more vulnerable to mutagens[45]. Mutational asymmetry also occurs on the transcribed and non-transcribed strands during transcription[46–48]. In

the B cell genome of mammals, localized RNA processing protein complex determines the strand-specific mutations, which catalyze proper antibody diversification[49,50]. In the genomes of tumors, transcription-coupled damage on the non-transcribed DNA strand and replication-coupled mutagenesis on the lagging-strand template have been detected, and these widespread asymmetric mutations have been proposed to potentially lead to cancer[51]. Our findings of strand-specific single-strand DNA breaks in chloroplast genome provide new insights into the molecular mechanisms of strand-specific mutations that occur in a broad range of diseases.

Previous findings confirm that ATH is a bona fide primase indispensable for plant organellar DNA synthesis resembling gp4 in bacteriophage T7, which is essential for the processive replication of phage DNA[14]. Knockout of Twinkle leads to depletion of mitochondrial DNA and lethality in humans and mice[12,52,53]. Our work also shows the null mutations of ATH lead to embryo lethality in Arabidopsis. These findings further confirmed the key roles of ATH in DNA replication and genome integrity maintenance of chloroplasts.

Dual localization in chloroplasts and mitochondria is a common feature of many organelle proteins encoded by the nucleus[23,54,55]. By subcellular localization analysis and immunoblots, we confirmed that the ATH protein also has bidirectional localization properties. Through further studies, we found that the R166K mutation in ATH can weaken R-loop accumulation and enhance genome stability in the chloroplast genome of the *atrnh1c* mutant. Since AtRNH1C is a protein specifically localized in chloroplasts, we speculate that the effect of ATH mutation is mainly present in chloroplasts. Immunoblot analysis showed that the expression levels of chloroplast proteins were recovered in *acs1* compared to *atrnh1c*, while there was no obvious difference in the expression of mitochondrial protein IDH (Supplementary Fig. 2b). Previous studies showed that DNA maintenance in mitochondria was mainly through the high levels of homologous recombination (HR)-based replication[56], and probably initially started the replication with RNA synthesized by RNA polymerase as the primer[57]. In addition, changing the dual localization ATH signal peptide with signal peptides that are specifically localized in chloroplasts could also rescue the phenotype of *acs1* (Supplementary Fig. 3d). Thus, the importance and function of ATH in mitochondria remain to be further investigated.

## Methods

### Plant growth and materials

All *Arabidopsis thaliana* materials used in this study are in the ecotype Columbia-0 (Col) background. The T-DNA insertion mutants *atrnh1c*, *rhon1*, and SALK_152246 were obtained from Nottingham Arabidopsis Stock Centre, UK. Surface-sterilized seeds were sown on 1/2 MS medium and incubated at 4 °C for 2 days for stratification. The plants were grown in the chamber under long-day conditions (day/night cycle of 16/8 h) at 22 °C in white light and 18 °C in the dark as described[20]. All plant materials are used from 21-day-old seedling leaves that grew on 1/2 MS medium, unless otherwise specifically indicated.

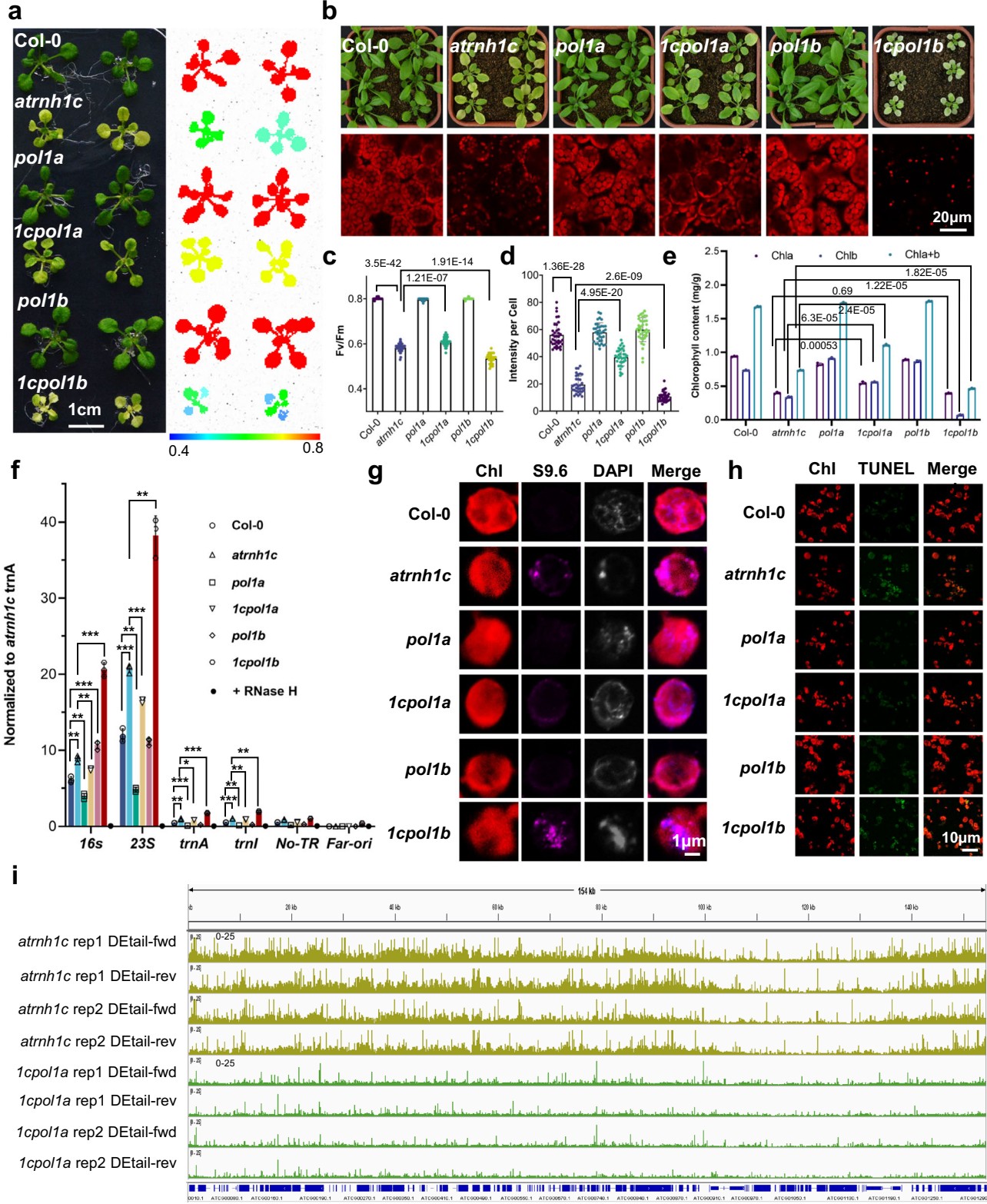

For the complementation experiments, *ATH* or *ATH(R166K)*, genomic DNA sequence from 1.5-kb upstream of ATG to 500-bp downstream of the stop codon were amplified and cloned into the binary vector pCambia1300, thus generating the *acs1^{ATH::ATH}* and *acs1^{ATH::ATH(R166K)}* vectors. Then the GFP or GUS tags were amplified and fused to the C-termini of ATH to generate the *acs1^{ATH::ATH-GFP/GUS}* and *acs1^{ATH::ATH(R166K) -GFP}* vectors. The vectors were constructed using the Fast-Cloning method, and the primers are listed in Supplementary

Data 1. Then the construct was transformed into *acs1* plants, and transformants were selected using hygromycin antibiotics.

To generate *ATH* overexpression transgenic plants, the coding sequence of the *ATH* gene without the stop codon was cloned into the binary vector pEarleyGate202 and fused with GFP or FLAG tags at the C-termini. Then the construct was transformed into Col-0 and *acs1* plants, and transformants were selected using hygromycin antibiotics. *AtRNH1C* overexpression transgenic plants were generated by cloning

**Fig. 7 | Mutation of Pol1A rescues the developmental defects of *atrnh1c* by restricting R-loop accumulation and maintaining genome integrity in chloroplasts. a** Photographs and chlorophyll fluorescence images of 14-day-old Col-0, *atrnh1c*, *pol1a*, *1cpol1a*, *pol1b*, and *1cpol1b* plants. Chlorophyll Fv/Fm values are presented in the lower panel. Scale bar, 1 cm. **b** Comparison of phenotype and chloroplasts from the leaves of 21-day-old Col-0, *atrnh1c*, *pol1a*, *1cpol1a*, *pol1b*, and *1cpol1b* plants. Chloroplasts are distinguished by chlorophyll autofluorescence (red). Scale bars, 20 μm. Experiments were repeated three times with similar results. **c** Chlorophyll Fv/Fm values of 21-day-old Col-0, *atrnh1c*, *pol1a*, *1cpol1a*, *pol1b*, and *1cpol1b* plants. $n = 32$. The graphs represent the mean ± SD. Statistical testing was performed by unpaired two-sided $t$ test. **d** The intensity of chlorophyll autofluorescence in leaf cells of different plants is indicated in (**b**). $n = 34$. The graphs represent the mean ± SD. Statistical testing was performed by unpaired two-sided $t$ test. **e** The chlorophyll contents of 21-day-old Col-0, *atrnh1c*, *pol1a*, *1cpol1a*, *pol1b*, and *1cpol1b* plant leaves. Three biological replicates were performed. The graphs represent the mean ± SD. Statistical testing was performed by unpaired two-sided $t$ test. **f** R-loop levels in the cp-rDNA head-on regions were detected by DRIP-qPCR. RNase H treated *atrnh1c* cpDNA sample was used as a negative control. Three biological replicates were performed. The graphs represent the mean ± SD. *$P < 0.05$; **$P < 0.01$ by unpaired two-sided $t$ test. **g** R-loop signal (magenta) accumulation in chloroplasts was detected by S9.6 immunostaining. Scale bar, 1 μm. Experiments were repeated three times with similar results. **h** TUNEL assays detect DNA damage. Scale bar, 10 μm. **i** Snapshots of DEtail-seq in the whole chloroplast genome of 21-day-old *atrnh1c* and *1cpol1a* plants. Experiments were repeated three times with similar results. Source data are provided as a Source Data file.

the coding sequence of *AtRNH1C* into the binary vector pEarley-Gate202 without the stop codon, fused with the HA tag at the C-terminus. Then the construct was transformed into Col-0 and *atrnh1c* plants, and transformants were selected using kanamycin antibiotics.

To generate the genomic mutation of the *ATH* and *Pol1A* gene, the plant CRISPR/Cas9 system was used as previously described[20,24]. The sequences of sgRNAs are listed in Supplementary Data 1.

All these vectors were transferred into Agrobacterium tumefaciens strain GV3101 and then transformed into Arabidopsis plants by the floral dip method. Transgenic lines were identified through selection using hygromycin or kanamycin antibiotics and verified by PCR and immunoblotting.

## Whole-genome sequencing-based mapping
The *atrnh1c* suppressor line *acs1* was backcrossed to *atrnh1c*, and the F2 population was grown on MS plates for 14-days. Two hundred seedlings with the green leaf phenotype were collected, and genomic DNA was extracted. The genomic DNA was then submitted to the DNA-sequencing facility for library preparation and sequencing on a NovaSeq 6000 system (Illumina) to generate 100-bp paired-end reads, yielding >20-fold genome coverage. The reads were mapped to a Col-0 reference genome (TAIR10), and the putative single-nucleotide polymorphisms (SNPs) were used as markers to identify regions with *atrnh1c* across the genome using SHOREmap software[22]. Only C/G-to-T/A transition (EMS-induced) SNP markers were further considered candidates. The causative mutation within the mapping interval was annotated using the SHOREmap software annotate function.

## Chlorophyll fluorescence measurements
Chlorophyll fluorescence was measured using the FluorCam (from the Institute of Botany, Chinese Academy of Sciences). The plants were first dark adapted for 30 min before measurement, and the minimum fluorescence yield (Fo) was measured with measuring light. A saturating pulse of white light was applied to measure the maximum fluorescence yield (Fm). The maximal photochemical efficiency of PSII was calculated based on the ratio of Fv (Fm-Fo) to Fm. For the image analysis, the corresponding data from the plants were normalized to a false-color scale with an assigned extremely high value of 0.8 (red) and a lower value of 0.4 (blue).

## Phylogenetic analysis and molecular modeling
The ATH protein sequences were used as queries to search against various species genomes in NCBI with BLASTP. Multiple sequences were submitted to the phylogenetic analysis tool NGPhylogeny.fr with default settings for sequence alignments. For phylogenetic tree construction, the FastME Output Tree was then uploaded to iTOL (version 5) for tree visualization. The 3D predicted structure of ATH was obtained from the AlphaFold Protein Structure Database (https://alphafold.ebi.ac.uk).

## Expression and purification of recombinant proteins
The full-length CDS of ATH, ATH(R166K), Pol1A, and Pol1B were amplified and cloned into the pGEX-4T vector and expressed in Rosetta (DE3) cells. Rosetta (DE3) cells were grown at 37 °C until the OD600 reached 0.6, and then 0.5 mM IPTG was added. The cells were then held at 18 °C and incubated overnight (16 h) with shaking. After centrifugation at 4000 g, the harvested cells were re-suspended in 1x PBS and sonicated on ice until the suspension became transparent. The supernatant was collected and incubated with GST agarose resins (YEASEN, 20507ES50) at 4 °C for 4 h. The agarose resins were washed four times with 1x PBS and the proteins were eluted with elution buffer (50 mM reduced glutathione in 1x PBS). The quality of GST-ATH and GST-ATHR166K proteins was tested by SDS-PAGE gel, and the protein concentrations were measured with a Bradford Protein Assay Kit (Beyotime, P0006C).

## Electrophoretic Mobility Shift Assay (EMSA)
Unlabeled and 3′-FAM-tagged synthetic oligonucleotides were used as probes (sequence: SGGASGGASGGASGGASGGASGGASGGA). The ATH and ATH(R166K) proteins (0.5 μg to 5 μg) were incubated with 20 pmol probes in 1x binding buffer (Beyotime, GS005) for 30 min, then the reaction mixture was separated on 8.5% native PAGE gel and visualized by Typhoon FLA9500.

## Template-directed primer synthesis and RNA-primed DNA synthesis
Primase reactions were assayed with 100 nM ssDNA template (5′-(T)$_7$GGGA(T)$_7$-3′), 100 μM GTP, CTP, UTP, and 10 μCi of [γ-$^{32}$P]-ATP (NEG502A) in a buffer containing 40 mM Tris−HCl pH 7.5, 50 mM potassium glutamate, 10 mM MgCl2 and 10 mM DTT. Each primase reaction contained varied amounts of recombinant protein as indicated in the figures. After incubation at 30 °C for 60 min, loading buffer (95% formamide, 0.1% xylene cyanol) was added to the reaction products, and then the products were separated on a 27% denaturing polyacrylamide gel containing 3 M urea. The autoradiographs were exposed to the phosphor screens for 2 days and then scanned using Typhoon FLA9500.

RNA-primed DNA synthesis were assayed with 100 nM ssDNA template (5′-(T)$_5$A(T)$_9$GGGGA(T)$_{10}$-3′), 100 μM NTPs, 100 μM dATP, 100 μM dGTP, 100 μM dTTP, and 10 μCi of [α-$^{32}$P]-dCTP (NEG513H) in a same buffer as above. Each primase reaction contained varied amounts of recombinant protein as indicated in the figures. After incubation at 30 °C for 60 min, the reaction products were detected as described above.

## Helicase assay
ATH helicase reactions were assayed in a buffer containing 10 mM Tris−HCl pH 8.0, 8 mM MgCl2, 1 mM DTT, and 5 mM ATP. 25 nM or 10 nM 3′-FAM-tagged dsDNA (SGGASGGASGGASGGASGGASGGAS GGA) were used as substrate. Each helicase reaction contained varied

amounts of recombinant protein as indicated in the figure. After incubation at 37 °C for 2 h, loading buffer (95% formamide, 0.1% xylene cyanol) was added to the reaction products, and then the products were separated on a 9% native polyacrylamide gel and scanned using Typhoon FLA9500.

## Protoplast transformation

The CDS of ATH and ATH(R166K) were cloned into pUC19-35S-eGFP. Protoplasts were extracted from 30-day-old leaves and transformed with plasmid-mediated by 20% PEG-Ca. After 16 h of transient expression, protoplasts were observed under fluorescence microscopy (Zeiss, LSM880).

## Chloroplast and mitochondrion fractionation

Chloroplast sub-fractionation analysis was performed as previously described[58] with minor modifications. Briefly, 21-day-old plants were homogenized in CIB buffer (10 mM HEPES−KOH pH 8.0, 150 mM sorbitol, 2.5 mM EDTA pH 8.0, 2.5 mM EGTA pH 8.0, 2.5 mM MgCl₂, 5 mM NaHCO₃, and 0.1% BSA) on ice. The homogenate was further filtered through a double layer of Miracloth and centrifuged at 200 g/4 °C for 3 min. Then the supernatant was transferred to new 50 ml tubes and centrifuged for 10 min at 1200 g/4 °C to obtain intact chloroplasts. The chloroplasts were resuspended in buffer II (0.33 M sorbitol, 5 mM MgCl₂, 2.5 mM EDTA pH 8.0, 20 mM HEPES−KOH pH 8.0) and buffer III (5 mM MgCl₂, 25 mM EDTA pH 8.0, 20 mM HEPES−KOH pH 8.0), successively. After centrifugation at 4 °C, the stroma was in the liquid supernatant, while the sediment contained the thylakoid fraction.

Mitochondrial fractionation was performed as previously described[55]. 21-day-old plants were homogenized in ice-cold grinding buffer (0.3 M sucrose, 25 mM tetrasodium pyrophosphate, 1% (w/v) polyvinylpyrrolidone-40, 2 mM EDTA, 10 mM KH₂PO₄, 1% (w/v) BSA, 20 mM sodium L-ascorbate, 1 mM DTT, 5 mM cysteine, pH 7.5). The homogenate was filtered through 4 layers of Miracloth and centrifuged at 2,500 g/4 °C for 5 min, and the supernatant was then centrifuged at 20,000 g/4 °C for 15 min. The pellet was resuspended in washing buffer (0.3 M sucrose, 10 mM TES, pH 7.5), and repeated 1500 g and 20,000 centrifugation steps. The resulting pellet was gently resuspended in washing buffer and fractionated on a Percoll step gradient (18% to 27% to 50%) by centrifugation at 40,000 g for 45 min. Mitochondria were collected at the 27% to 50% interface and diluted with washing buffer. After centrifugation at 31,000 g/4 °C for 15 min, the mitochondrial pellet was collected for use.

## Protein extraction and immunoblot analysis

Total protein was extracted with protein extraction buffer (50 mM Tris-HCl pH 7.4, 154 mM NaCl, 10% glycerol, 5 mM MgCl₂, 1% Triton X-100, 0.3% NP-40, 5 mM DTT, 1 mM PMSF, and protease inhibitor cocktail). Anti-FLAG (Sigma, F1804), anti-GFP (ABclonal, AE012), anti-HA (Beyotime, AF5057), anti-plant-actin (ABclonal, AC009), anti-RPOB (PhytoAB, PHY1701), anti-PetA (PhytoAB, PHY0023), anti-RbcL (Agrisera, AS03037A), anti-PsaA (PhytoAB, PHY0053A), and anti-IDH (PhytoAB, PHY0098A) were used as primary antibodies, and goat anti-mouse (EASYBIO, BE0102) or goat anti-rabbit antibodies (EASYBIO, BE0101) were used as secondary antibodies.

## GUS staining

The *acs1^ATH::ATH-GUS* transformed Arabidopsis plants were incubated with staining buffer (0.1 M K₃[Fe(CN)₆], 0.1 M K₄[Fe(CN)₆], 1 M NaH₂PO₄, 1 M Na₂HPO₄, 0.5 M EDTA (pH 8.0) and 20% methanol) under vacuum for 10 min and then overnight at 37 °C. After staining, the plants were washed three times with 100% alcohol and then photographed.

## Seed clearing and observation

Seed clearing was performed as previously described[55,59] with minor modifications. In brief, developing siliques at various stages were fixed in ethanol:acetic acid (9:1) and washed with 70% ethanol. The seeds were isolated, mounted on slides in Hoyer's medium (glycerol/water/chloral hydrate in a ratio of 1:2:8, v/v/w), and observed under a differential interference contrast (DIC) light microscope (Olympus, BX53).

## Chloroplast isolation and chloroplast DNA extraction

Leaves of 21-day-old plants were harvested to extract intact chloroplasts using a chloroplast isolation kit (Invent, CP-011) following the instructions from the manufacturer. Briefly, plant leaves were added to a filter column containing 200 µl cold buffer A, and were gently ground with a grinding rod for 2 min on ice. The filter was capped and centrifuged at 2000 g for 5 min. The pellet was suspended in 200 µl cold buffer B and centrifuged at 2000 g for 10 min. The chloroplast was washed twice in CIB buffer before use. The chloroplasts were used for the comet assay, TUNEL assay, and immunofluorescence staining.

For PFGE, Slot-blot, DRIP, and cpChIP, chloroplasts were extracted by grounding plant leaves in 20 ml ice-cold CIB buffer (10 mM HEPES−KOH (pH 8.0), 150 mM sorbitol, 2.5 mM EDTA (pH 8.0), 2.5 mM EGTA (pH 8.0), 2.5 mM MgCl₂, 5 mM NaHCO₃, and 0.1% BSA). The homogenate was filtrated through two layers of Miracloth and centrifuged at 200 g/4 °C for 3 min. Then the supernatant was transferred to new 50 ml tubes and centrifuged for 10 min at 1200 g/4 °C. The pellet was then suspended with cold CIB buffer and was ready for use.

For chloroplast DNA extraction, chloroplasts were lysed in chloroplast DNA extraction buffer (CIB with 1% SDS and proteinase K) at 37 °C overnight with shaking, and then SDS was removed by adding 20 µM KAc. Chloroplast DNA was purified by phenol/chloroform/isoamyl alcohol (25:24:1, v/v/v) and precipitated with an equal volume of isopropyl alcohol at -20 °C overnight. The chloroplast DNA was dissolved in 1x TE.

## DAPI staining and immunostaining

For DAPI staining, the intact chloroplast was fixed in 4% paraformaldehyde for 10 min at room temperature. Then the chloroplast was washed three times with 1x PBS and stained with DAPI. The stained chloroplast was observed by a confocal microscope (Zeiss, LSM880).

For immunostaining, the fixed chloroplasts were refixed on PLL-coated slides (CITOGLAS, 188105) with 4% paraformaldehyde for 10 min and then washed three times with 1x PBS. Then the chloroplasts were pretreated with RNase III for 30 min and washed with 1x PBS. The samples were then blocked with blocking buffer (1% BSA, 0.3% Triton X-100 in 1x PBS) for 20 min at room temperature and incubated with 100 µl of S9.6 antibody diluted in blocking solution (1:100) at 4 °C overnight. The slides were washed three times with 1x PBS and incubated with 100ul of secondary antibody for 1 h at room temperature. Then, the cells were washed three times washing with 1x PBS. 20ul of protective agents (Southern Biotech, 0100-20) were added to the slides and covered with cover glass. The observation was performed by a confocal microscope (Zeiss, LSM880).

## Slot-blot hybridization analysis

The detailed Slot-Blot assay was described previously[20] with minor modifications. Briefly, 5 µg chloroplast DNA extracted from different samples was treated with 1 U of RNase III (NEB, M0245S) at 37 °C for 30 min, then purified with phenol/chloroform/isoamyl alcohol (25:24:1, v/v/v) and precipitated with an equal volume of isopropyl alcohol. The purified DNA was spotted on Hybond N+ membrane using a slot-blot apparatus and vacuum suction. The membrane was then crosslinked and blocked in 5% milk-TBST, and detected by S9.6 antibody (DNA:RNA hybrid-specific antibody).

## Chloroplast chromatin immunoprecipitation (cpChIP)

cpChIP was performed as described previously[36]. The chloroplasts were cross-linked with 1% formaldehyde for 10 min, and the cross-linking reaction was stopped by adding 150 µl 1 M glycine and

incubating for 10 min. Then the cross-linked chloroplasts were washed twice with CIB and lysed in lysis buffer (50 mM Tris-HCl (pH 7.6), 0.15 M NaCl, 1 mM EDTA (pH 8.0), 1% Triton X-100, 0.1% SDS, 0.1% sodium deoxycholate). Chloroplast DNA was sheared with sonication into fragments of ~500-bp. The supernatant was incubated with an anti-GFP antibody (abcam, ab290) overnight at 4 °C. Plants without a GFP tag were adopted as the negative control. ChIP-qPCR was performed using the immunoprecipitated DNA and input DNA. Primers corresponding to four rDNA regions were used for detection and are listed in Supplementary Data 1.

## DNA:RNA hybrid immunoprecipitation (DRIP)

5 μg of chloroplast DNA was fragmented with 5 U of DdeI (NEB, R0175V), MseI (NEB, R0525S), RsaI (NEB, R0167V), and AluI (NEB, R0137V) at 37 °C for 12 h. RNase H pretreated *atrnh1c* cpDNA was used as the negative control. The fragmented DNA was then purified by phenol/chloroform/isoamyl alcohol (25:24:1, v/v/v) and precipitated with an equal volume of isopropyl alcohol. 2 μg of purified fragment DNA with or without RNase H treatment was incubated with 10 μg of S9.6 antibody overnight at 4 °C. Samples were further incubated with 50 μl Protein G beads (Invitrogen, 10004D) for 4 h at 4 °C. The immunoprecipitated DNA was purified as mentioned above. The primers used for DRIP-qPCR are listed in Supplementary Data 1.

## DEtail-seq

DEtail-seq assay was described previously[29] with minor modifications. Briefly, the chloroplasts were embedded in low-melting-point agarose and lysed in lysis buffer as described above. Agarose plugs were then washed in 1x TE buffer 4 times at 37 °C, with the first two washes containing 1 mM PMSF. Then, agarose plugs were cut into small pieces and lysed in 10 μg/ml RNase A at 37 °C overnight. The lysed agarose pieces were then washed in 1x TE buffer 5 times at 37 °C, followed by two washes with 300 μl 1x CutSmart (NEB, B7204S) at 37 °C. The agarose pieces were then incubated with 3 μl I-CeuI (NEB, R0699S) in 150 μl 1x CutSmart Buffer at 37 °C for 12 h and washed 3 times with 1x TE buffer. Then the T7 ligation process was conducted by adding 65 μl (T7 Buffer 8 μl, T7 Adapter 5 μl, T7 Enzyme Mix II 6 μl, Low-EDTA TE 46 μl) T7 Tailing & Ligation solution (ABclonal, RK20228) and incubating at 37 °C for 12 h. The DNA in agarose was purified using a DNA gel purification kit (Magen, D2111-02), and fragmented to ~250 bp using a focused ultra-sonicator (Covaris, S220). The following library preparation steps were conducted according to the manual (ABclonal, RK20228).

## In vitro removal of RNA primer by AtRNH1C

The assay was performed as previously described[55]. 100 nM FAM-labeled RNA:DNA hybrid (8-bp) was used as the substrate. The reaction was performed in a buffer containing 50 mM KCl, 4 mM MgCl2, 20 mM HEPES-KOH pH 7.0, 4% glycerol, 50 μg/ml BSA, and 1 mM DTT. After incubation with 100 to 400 nM purified GST-AtRNH1C or GST-GFP proteins for 30 min, the reactions were stopped by 20 mM EDTA. The products were separated on a 12% native polyacrylamide gel and then scanned using Typhoon FLA9500.

## Single-cell gel electrophoresis assay (Comet assay)

The comet assay was performed as previously described[20]. Briefly, 10 μl of intact chloroplasts were mixed with 90 μl of LM Agarose at 37 °C, and 50 μl of each sample was added to the CometSlide. Slides were incubated at 4 °C in the dark for 20 min for solidification and then incubated in lysis solution (R&D, 4250-050-01) overnight at 4 °C in the dark. The slides were further incubated in neutral electrophoresis buffer (50 mM Tris and 150 mM sodium acetate, pH 9.0) for 30 min. For electrophoresis, the slides were run at 1 V/cm for 20 min in neutral electrophoresis buffer. Slides were then incubated in DNA precipitation solution (1 M NH$_4$Ac in 95% ethanol) for 30 min and 70% ethanol for another 30 min. After drying at 37 °C, the slides were stained with

SYBR Green for 30 min followed by a water wash. Samples were visualized by epifluorescence microscopy (Olympus, BX53) at 488 nm. The OpenComet tool[60] launched from ImageJ software was used for the analysis and quantification of the results.

## Terminal deoxynucleotidyl transferase dUTP nick end labeling (TUNEL assay)

TUNEL assay was performed using a TUNEL Apoptosis Assay Kit-FITC (7sea, AT005-1) according to the manufacturer's instructions. Briefly, the intact chloroplasts were extracted and fixed onto PLL-coated slides. Then the samples were blocked by blocking buffer (1% BSA, 0.3% Triton X-100 in 1x PBS) for 20 min at room temperature. The slides were rinsed with 1x PBS three times and then incubated with TdT reaction mixture for 1 h at 37 °C. After three washes with 1x PBS, chlorophyll autofluorescence and TUNEL fluorescence were captured under confocal microscopy (Zeiss, LSM880). The fluorescence intensity was quantified by ImageJ software.

## EdU labeling of Arabidopsis chloroplast

21-day-old plants were transferred to liquid 1/2 MS medium with 20 μM EdU (BeyoClick™ EdU Cell Proliferation Kit with Alexa Fluor 488) and grown in the chamber for 17 h. Then the plants were washed with 1x PBS three times and fixed in 4% paraformaldehyde in 1x PBS for 10 min. After being fixed, the plants were washed twice with 1x PBS, and a MinuteTM Chloroplast Isolation Kit (CP-011, Invent) was used to extract intact chloroplasts. The chloroplasts were fixed in 4% paraformaldehyde again on PLL-coated slides for 30 min and covered with 0.3% Triton X-100 in 1x PBS for 20 min at room temperature. Then the slides were incubated with 50 μl Click-iT® reaction cocktail (43 μl of 1x Click-iT® EdU reaction buffer, 2 μl of CuSO$_4$, 0.1 μl of Alexa Fluor® azide, and 5 μl of 1XClick-iT® EdU buffer additive) for 1 h at room temperature. Then, the cells were washed three times with 1x PBS, and 10 μl protective agent (Southern Biotech, 0100-20) covered with cover glass. The observation was performed by a confocal microscope (Zeiss, LSM880).

## Two-dimensional gel electrophoresis of replication intermediates (2D-gel)

Two-dimensional gel electrophoresis was performed as described previously[20]. A total of 20 μg cpDNA from each sample was digested with 10 U of the restriction enzymes (AseI and BglI) and precipitated with isopropanol. For the first-dimension gel electrophoresis, the digested DNA was loaded onto a 0.3% agarose gel without ethidium bromide, in 0.5x TBE buffer at 0.7 V/cm for 30 h. The second-dimension gel electrophoresis was performed in a 1% agarose gel containing 0.3 μg/ml ethidium bromide, at 6 V/cm for 5 h at 4 °C. DNA was transferred onto Hybond N+ membrane (GE, RPN303B) according to standard DNA gel blotting methods. The blots were hybridized to radiolabeled probes labeled with [α-$^{32}$P]-dCTP (NEG513H) using a Random Primer DNA Labeling Kit Ver. 2 (Takara, 6045). The autoradiographs were exposed to the phosphor screens for 10 days and scanned by Typhoon FLA9500.

## Pulsed-field gel electrophoresis (PFGE)

The chloroplasts suspended in CIB were mixed with 1% low-melting point agarose (Promega, V2111) dissolved in TE buffer (1:1, v/v) at 37 °C. The plugs were solidified at 4 °C for 30 min and then lysed in lysis buffer (1% sarkosyl, 0.45 M EDTA, 10 mM Tris-HCl (pH 8.0) and 2 mg/mL proteinase K) at 48 °C for 16 h with shaking. The lysis buffer was exchanged three times. Agarose plugs were then washed in 1x TE buffer 6 times at 4 °C, with the first two washes containing 1 mM PMSF, filled into 1% agarose gel, and subjected to electrophoresis in 0.5x TBE for 42 h at 14 °C using a CHEF Mapper XA system (Bio-Rad). A Lambda Ladder (New England Biolabs; N0341) was used to indicate the molecular weight. The detailed electrophoresis parameters were 5 to 120 s of pulse time at 4.5 V/cm. After EtdBr staining and photography, the gel was blotted onto

a Hybond N+ membrane (GE, RPN303B) according to standard DNA gel blotting methods. A 505-bp fragment of the chloroplast rbcL gene (55677–56181) was labeled with [α-$^{32}$P]-dCTP (NEG513H) using Random Primer DNA Labeling Kit Ver. 2 (Takara, 6045) and used as a probe for hybridization. The autoradiographs were exposed to the phosphor screens for 5 days and then scanned using Typhoon FLA9500.

## Reporting summary

Further information on research design is available in the Nature Portfolio Reporting Summary linked to this article.

## Data availability

The sequencing data generated in this study have been deposited into NCBI's Gene Expression Omnibus (GEO) database and are accessible through the GEO Series accession number GSE215443. Source data are provided with this paper.

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

## Acknowledgements

The authors thank all the members of The Sun Lab and Professor Jie Ren (from Beijing Institute of Genomics, Chinese Academy of Sciences) for their helpful discussions and constructive suggestions. We thank Mrs. Dan Zhang and Mrs. Fang Liu (from the Center of Biomedical Analysis, Tsinghua University) for their assistance with confocal observation and Bioanalyzer analysis, respectively. We thank Mrs. Yan Yin (from the Institute of Botany, Chinese Academy of Sciences) for assisting with the measurement of chlorophyll fluorescence using FluorCam. This work was funded by grants from the National Natural Science Foundation of China (grants 32261133529 and 32170321 to Q. Sun, and 32070651 to W. Zhang). The Sun Lab is supported by the Tsinghua-Peking Center for Life Sciences. W. Zhang is supported by the China Postdoctoral Science Foundation Project (2019M660610) and the postdoctoral fellowship from Tsinghua-Peking Center for Life Sciences.

## Author contributions

Q.S. conceived the study and designed the experiments with W.Z.; Z.Y. conducted the EMS mutagenesis screen of atrnh1c suppressors; W.W. assisted W.Z. in the DEtail-seq and PFGE assays; W.Z. performed the rest of the experiments. W.Z. and Q.S. wrote the manuscript, and all authors read and approved the final manuscript.

## Competing interests

The authors declare no competing interests.
