## [Peer Review File · Nature Communications]

Primase excites the competition between transcription and replication on the same template strand to boost DNA damageREVIEWER COMMENTS

Reviewer #1 (Remarks to the Author):

Zhang, Sun et al, Nature Communications (2023)

Collisions of DNA replication complex and transcription complex induces R-loops (TRC-transcription replication collision) and leads to various DNA damage events. Specially, transcription units (often genic sequences) when oriented in the head-on (HO) direction with replication start sites- undergo head-on TRC (HO-TRC) leading to genomic instability and associated DNA mutagenesis. Previous studies from the Sun laboratory have shown that loss of RNase H1 subunit (RNase H1C) leads to increased R-loop accumulation in the chloroplast genome and associated genomic instability. Seeking rescue mutants of R-loop associated genomic instability, the authors have discovered that chloroplast genome replication primase ATH promotes R-loop accumulation. Using various biochemical, genomic and genetic approaches, the manuscript attempts to establish a firm role of the ATH protein in promoting nucleic acid hybrid formation and replication control. I found the study of interest and, below I suggest some revisions that the authors can use to improve the manuscript.

1. For Fig. 1E, the authors should demonstrate in the same figure that the Protein levels loaded for both the GST-ATH and GST-ATHR166K mutants. The protein levels loaded for each dilution is crucial since the difference of ssDNA binding ability of the two proteins are detectable but not absolutely different. Small differences in protein amounts or quality can induce differences in data presented.
2. In fig. 4A, the *acs1* mutant has lower S9.6 signals than the *atrnh1*, but the levels in Col-0 is even lower. Thus, the primase mutations do not completely rescue the hybrid accumulation. First, the RnaseH treatment should be done for all the conditions/mutants shown in the slot blot. Second, some other mutants that accumulate DNA/RNA hybrids should be provided here as positive control (and for comparison). For example, recent studies have shown exosome mutations and THO complex mutations accumulate DNA/RNA hybrids- can that be done along with the experimental lanes in this slot blot.
3. For Figure 5A-C, the strand specific DRIP-seq can have contamination of RNA/RNA hybrids. Thus, the presentation of the Rnase H treated control of fwd/rev DRIP-seq are important to demonstrate.
4. In Fig. 1K, the authors show S9.6 staining for hybrids. This experiment is concerning since there has been reports about S9.6 may not be optimal for immuno-staining. Can authors consider using a dead RNaseH-GFP staining for this experiment.
5. Finally, authors discuss strand specific mutation mechanisms in his manuscript. However, the relevance of this observations in context of mammalian biology could be better discussed. Somatic strand specific mutations are useful in the immunological system for antibody diversity (PMID: 28431250, 33526923) but contrastingly asymmetric mutations are seen in cancer genomes and have been proposed to be an oncogenic event (PMID: 26806129). Some of these previous studies that set the field of strand specific mutations and relevance to transcription and replication should be discussed. This will make the study of significantly higher general interest.

Reviewer #2 (Remarks to the Author):

The manuscript by Zhang et al., identified a mutation in chloroplast-localized primase ATH, which can rescue the phenotype of *atrnh1c* mutant, including weakening the binding affinity of DNA template, reducing the activities of RNA primer synthesis and delivery, slowing down DNA replication and relieving the competition of transcription-replication. Overexpression of ATH boosts HO-TRCs and exacerbates DNA damage. Their results are very interesting, and most results and evidence presented in this study generally seem to support the conclusion. However, there are a few concerns need to be addressed:

Major concerns:

1. What is the effect of R166 of ZBD domain in ATH on the helicase function of ATH? Does it affect the activity of primase?
2. The phenotypes of the two homozygous mutants of ATH and RNH1C are inconsistent. When and how do they work together? Under certain circumstances? Or is it constitutive regulation?
3. A more detailed analysis of ath+/- plants is important to understanding the role of ATH in the genome integrity.
4. The authors also pointed that mutation of Pol1A can rescue the growth defects of atrnh1c by restricting HO-TRCs in chloroplasts. Given that Pol1A and ATH both play a role in DNA replication and rescue the growth defect of atrnh1c,
What is the relationship among the ATH, RNH1C and Pol1?
5. Fig 7A, compared with atrnh1c, the growth defects of the 1cpol1a double mutant were partially restored, but 1cpol1b appears smaller than atrnh1c. How to explain this difference of phenotype?

Minor points:

1. Fig 1A and 1B, a brief description of acs1ATH::ATH and acs1ATH::ATH-GFP plants should be included in the text.
2. Fig3C, the size of the marker band is not marked in the figure or labeled in the figure legends.
3. Fig 4B and Fig 7E, significant differences were not marked clearly enough.

We are grateful to the Reviewers for their constructive comments, which have allowed us to improve our manuscript. Detailed point-by-point responses addressing the reviewers' comments and queries are described below.

Reviewer #1's questions and our responses:

Zhang, Sun et al, Nature Communications (2023)

Collisions of DNA replication complex and transcription complex induces R-loops (TRC-transcription replication collision) and leads to various DNA damage events. Specially, transcription units (often genic sequences) when oriented in the head-on (HO) direction with replication start sites- undergo head-on TRC (HO-TRC) leading to genomic instability and associated DNA mutagenesis. Previous studies from the Sun laboratory have shown that loss of RNase H1 subunit (RNase H1C) leads to increased R-loop accumulation in the chloroplast genome and associated genomic instability. Seeking rescue mutants of R-loop associated genomic instability, the authors have discovered that chloroplast genome replication primase ATH promotes R-loop accumulation. Using various biochemical, genomic and genetic approaches, the manuscript attempts to establish a firm role of the ATH protein in promoting nucleic acid hybrid formation and replication control. I found the study of interest and, below I suggest some revisions that the authors can use to improve the manuscript.

Thank you very much for your supportive comments.

1. For Fig. 1E, the authors should demonstrate in the same figure that the Protein levels loaded for both the GST-ATH and GST-ATHR166K mutants. The protein levels loaded for each dilution is crucial since the difference of ssDNA binding ability of the two proteins are detectable but not absolutely different. Small differences in protein amounts or quality can induce differences in data presented.

Thank you for your suggestions. We have followed your suggestions and presented the expression and purification of GST-ATH and GST-ATHR166K proteins, and SDS-PAGE analysis showed that both proteins have high quality after purification (As shown below, **Response Figure 1**, now included in **Fig. S2F**).

Response Figure 1, SDS-PAGE analysis of expression and purification of GST-ATH and GST-ATHR166K proteins. lane 1: cell lysate before induction, lane 2: cell lysate after 16 h of induction by 0.5 mM IPTG at 18°C, lane 3: supernatant after sonication and centrifugation, lane 4: purified protein eluted by 50 mM reduced glutathione. The predicted size of GST-ATH proteins was indicated by arrows.

For your concern about the comparison of the protein levels loaded, we measured the protein concentration with a Bradford Protein Assay Kit (Beyotime, P0006C), and equal amounts of the two proteins were used for further assays. We have added the statement of equal amounts of the two proteins in the figure legends of Fig. 1E.

We have added more details of protein quality and concentration detection to the Methods section (**Lines 437-439**): “The quality of GST-ATH and GST-ATHR166K proteins was tested by SDS-PAGE gel (Figure S2F). The protein concentrations were measured by a Bradford Protein Assay Kit (Beyotime, P0006C), and equal amounts of the two proteins were used for further assays.”

2. In fig. 4A, the *acs1* mutant has lower S9.6 signals than the *atrnh1*, but the levels in Col-0 is even lower. Thus, the primase mutations do not completely rescue the hybrid accumulation. First, the RnaseH treatment should be done for all the conditions/mutants shown in the slot blot. Second, some other mutants that accumulate DNA/RNA hybrids should be provided here as positive control (and for comparison). For example, recent studies have shown exosome mutations and THO complex mutations accumulate DNA/RNA hybrids- can that be done along with the experimental lanes in this slot blot.

Thank you for underlining this deficiency. We have followed your advice and added the RNase H treatment control of all the slot blot assays (As shown below, **Response Figures 2-4**, now included in **Fig. 4**, **Fig. 6**, and **Fig. S11**, respectively).

Response Figure 2, Slot-blot assays of the overall chloroplast R-loop levels of 21-day-old Col-0, *atrnh1c*, *acs1*, *acs1^{ATH::ATH}* and *acs1^{ATH::ATH-GFP}* plants. RNase H-treated DNA was used as a negative control. R-loops were detected using the RNA:DNA hybrid antibody S9.6 (up panel). DNA loading was stained using SYBR Green (down panel).

Response Figure 3, Slot-blot assays of the overall chloroplast R-loop levels of 21-day-old Col-0, *Col-0^{35S::ATH-FLAG}*, *acs1*, and *acs1^{35S::ATH-GFP}* plants. RNase H-treated DNA was used as the negative control. R-loops were detected using the RNA:DNA hybrid antibody S9.6 (up panel). DNA loading was stained using SYBR Green (down panel).

Response Figure 4, Slot-blot assays of the overall chloroplast R-loop levels of 21-day-old *acs1*, *1crhon1*, and *rhon1acs1* plants. RNase H-treated DNA was used as the negative control. R-loops were detected using the RNA:DNA hybrid antibody S9.6 (up panel). DNA loading was stained using SYBR Green (down panel).

We also followed your suggestion and looked for other mutants that accumulate DNA/RNA hybrids for comparison. In our previous study, we conducted a screen of chloroplast R-loop regulators, with known transcription regulation and RNA processing activity in the chloroplast (Yang et al., 2020, PMID: 31914390). By S9.6 slot-blot hybridization analysis of mutants that carried defects in chloroplast development, we didn't find the increase of R-loop levels compared to Col-0 in tested mutants except for *atrnh1c* and *rhon1* (As shown below, **Response Figure 5**, from Figure S2 of Yang et al., 2020).

Response Figure 5, Slot blot hybridization analysis of global chloroplast R-loop levels of both Col-0 and other T-DNA mutants. Serial dilution of chloroplast DNAs (100, 200, 400 ng) extracted

from Col-0, *atrnh1c*, and other mutants with or without RNase H treatment was slotted onto a nylon membrane and detected using the DNA:RNA hybrid antibody S9.6 (data from Yang et al., 2020, PMID: 31914390).

Since no exosome or THO complex factors have been reported in the semiautonomous organelle chloroplast, we conducted S9.6 slot-blot assays of the nuclear THO complex mutants *tho1* and *tho6* in Arabidopsis (**Response Figure 6**). We could see the total amount of R-loops increase in the *tho* mutants. However, this is not relevant to the regulation of chloroplast R-loops. We will use these mutants to characterize the function of the THO complex in the regulation of nuclear R-loops in the future.

Response Figure 6, Slot blot hybridization analysis of global R-loop levels in Col-0, *tho1*, and *tho6* mutants. RNase H-treated DNA was used as the negative control. R-loops were detected using the RNA:DNA hybrid antibody S9.6 (up panel). DNA loading was stained using SYBR Green (down panel).

In our manuscript, we also used *1crhon1*, the double mutant of *atrnh1c* and *rhon1*, in the S9.6 slot-blot assays (Figure S11E), which accumulates more R-loops in chloroplasts than *atrnh1c* single mutant and can be regarded as a positive control.

3. For Figure 5A-C, the strand specific DRIP-seq can have contamination of RNA/RNA hybrids. Thus, the presentation of the Rnase H treated control of fwd/rev DRIP-seq are important to demonstrate.

Thank you for your comments. We have added the RNase H treated *atrnh1c* of fwd/rev DRIP-

seq in **Figures 5A-C** (As shown in below, **Response Figure 7**).

Response Figure 7, HO-TRCs cause single-strand DNA breaks at the end of the transcription units. (A) Snapshots of ssDRIP-seq, DETAIl-seq, and ATH ChIP-seq in the whole genome of the chloroplast. The green, blue, and pink sticks show a large single-copy (LSC) region, a small single-copy (SSC) region, and two inverted repeats (IR) regions, respectively. The red and blue ellipses show OriA and OriB. One of two replicates is shown. (B) Snapshots of ssDRIP-seq, DETAIl-seq, and ATH ChIP-seq in the rDNA region of the chloroplast indicated by the overline in (A). (C) Snapshots of ssDRIP-seq and DETAIl-seq in the area indicated by the overline in (B).

4. In Fig. 1K, the authors show S9.6 staining for hybrids. This experiment is concerning since there has been reports about S9.6 may not be optimal for immuno-staining. Can authors consider using a dead RNasH-GFP staining for this experiment.

Thank you for your opinion. For Fig. 1K, we think you may mean Fig. 6K here. As you mentioned, it has been previously reported that S9.6 may not be optimal for immuno-staining to detect RNA:DNA hybrid due to possible interference from dsRNA. However, binding affinity measurements in new studies showed S9.6 exhibits specificity for DNA-RNA hybrid over

dsRNA (PMID: 35347133, PMID: 35550870), and our previous immuno-staining assays in chloroplast also showed dsRNA does not affect the R-loop recognitions specificity of S9.6 antibody (Yang et al., 2020, PMID: 31914390, see the **Response Figure 8** shown below).

Response Figure 8, dsRNA does not affect the R-loop recognition specificity of S9.6 antibody. S9.6 and anti-dsRNA immunostaining was performed in the chloroplast of *rhon1/atrnh1c* double mutant with RNase H or RNase III pretreatment (data from Yang et al., 2020, PMID: 31914390).

Moreover, to avoid the effects of dsRNA signals on the results of RNA:DNA hybrids by S9.6 immunostaining, the chloroplasts were pretreated with RNase III for 30 minutes, then blocked and incubated with S9.6 antibody. The detailed experimental procedures of chloroplast S9.6 immunostaining have been described in the **Methods (Lines 533-544)**.

In addition, following your suggestions, we used a dead AtRNH1C(D222N)-mCherry to co-transformed with ATH-GFP in Col-0 protoplast, and the co-localization of the two proteins can also be seen in the nucleoids (As shown in below, **Response Figure 9**, now included in **Fig. S9H**). We have added the corresponding content to the **Results (Lines 258-261)**.

Response Figure 9, Co-localization of ATH-GFP and AtRNH1C(D222N)-mCherry (mCherry) in Col-0 protoplast. The panels beneath are the magnified images of the boxed areas; scale bars, 5 µm.

5. Finally, authors discuss strand specific mutation mechanisms in his manuscript. However, the relevance of this observations in context of mammalian biology could be better discussed. Somatic strand specific mutations are useful in the immunological system for antibody diversity (PMID: 28431250, 33526923) but contrastingly asymmetric mutations are seen in cancer genomes and have been proposed to be an oncogenic event (PMID: 26806129). Some of these previous studies that set the field of strand specific mutations and relevance to transcription and replication should be discussed. This will make the study of significantly higher general interest.

We sincerely appreciate your valuable comments. We have followed your suggestions and discussed strand specific mutation mechanisms in the context of mammalian biology as follows (**Lines 340-351**): “Strand-specific mutations occur in the process of DNA replication and transcription. Discontinuous synthesis of the lagging strand during replication produces a series of Okazaki fragments, the 5’ ends of which have increased levels of nucleotide substitution (PMID: 25624100). Additionally, longer exposure as ssDNA may cause the lagging strand to be more vulnerable to mutagens (PMID: 25449133). Mutational asymmetry also occurs on the transcribed and non-transcribed strands during transcription (PMID: 25251854, PMID: 32332764, PMID 20565875). In the B cell genome of mammals, localized RNA processing protein complex determines the strand-specific mutations, which catalyze proper antibody diversification (PMID: 28431250, 33526923). In the genomes of tumors, transcription-coupled damage on the non-transcribed DNA strand and replication-coupled mutagenesis on the lagging-strand template have been detected, and these widespread asymmetric mutations have been proposed to potentially lead to cancer (PMID: 26806129). Our findings of strand-specific DNA breaks in chloroplast genome provide new insights into the molecular mechanisms of strand-specific mutations that occur in a broad range of diseases.”

Reviewer #2's questions and our responses:

The manuscript by Zhang et al., identified a mutation in chloroplast-localized primase ATH, which can rescue the phenotype of atrnh1c mutant, including weakening the binding affinity of DNA template, reducing the activities of RNA primer synthesis and delivery, slowing down DNA replication and relieving the competition of transcription-replication. Overexpression of ATH boosts HO-TRCs and exacerbates DNA damage. Their results are very interesting, and most results and evidence presented in this study generally seem to support the conclusion.

We appreciate very much your positive feedback for our manuscript.

However, there are a few concerns need to be addressed:

Major concerns:

1. What is the effect of R166 of ZBD domain in ATH on the helicase function of ATH? Does it affect the activity of primase?

Thank you for your constructive suggestions. We compared the helicase activities of the wild type protein ATH and the mutant ATHR166K protein, and found that the R166K mutation could affect the helicase activity of ATH (As shown below, **Response Figure 10**, now included in **Fig. S2I**). We hypothesize that this is due to the weakening of the protein's ability to bind to the DNA template. We have added the detailed procedures of helicase assay to the **Methods (Lines 460-466)**, and added the relevant content to the **Results (Lines 130-132)**.

Response Figure 10, Helicase assay of the GST-ATH and GST-ATHR166K proteins. 25 nM (up panel) and 10 nM (down panel) 3'-FAM-tagged dsDNA was used as substrate, respectively. Both proteins were loaded from 100 to 400 nM as indicated.

2. The phenotypes of the two homozygous mutants of ATH and RNH1C are inconsistent. When and how do they work together? Under certain circumstances? Or is it constitutive regulation? We are sorry for the unclear explanation. In our previous study, we revealed that the knockout mutation of AtRNH1C (SAIL_97_E11) showed obvious dwarf and yellowish (pale-green) leaf phenotypes (Yang et al., 2017, PMID: 28939594). In this work, we found the homozygous null mutants of ATH lead to embryo lethality (Figure 3), and a point mutation in the ZBD domain of ATH (R166K) rescues the developmental defects of *atrnh1c* by reducing the activities of RNA primer synthesis during DNA replication. Furthermore, over-expression of ATH can lead to more R-loops in the chloroplast genome and yellowish phenotypes resembling *atrnh1c* mutant (Figure 6), and the growth defects caused by ATH over-expression can be rescued by over-expression of AtRNH1C synchronously (Figure S10). Based on these results, we hypothesize that in the chloroplast genome of wild type plants, ATH and AtRNH1C both ensure the proper DNA replication and avoid genomic instability.

3. A more detailed analysis of *ath*^{+/-} plants is important to understanding the role of ATH in the genome integrity.

Thank you for your valuable advice. We have followed your suggestions and carefully analyzed the phenotypes of the CRISPR/Cas9-generated *ath*^{+/-} and *1cath*^{+/-} heterozygous plants, and the *1cath-3*^{+/-} plants displayed partially rescued yellowish phenotypes compared to the *atrnh1c* mutant, while the phenotype of *1cath-5*^{+/-} plants recovers slightly (As shown in below, **Response Figure 11**, now included in **Fig. S6F**). The recovery could be because of loss of ATH protein levels, as the heterozygous mutants carry only one copy of the functional *ATH* gene (Fig. S6G). Furthermore, by S9.6 slot-blot assay, we found a significant decrease of R-loops in the chloroplast of *1cath*^{+/-} plants compared to that of *atrnh1c* (Fig. S6H). We have added the relevant content to the **Results (Lines 177-182)**.

Response Figure 11, (F) Phenotype of 14-day-old Col-0, *atrnh1c*, and heterozygous ATH mutants. **(G)** Schematic representations of the ATH proteins caused by CRISPR/Cas9-generated *ath-3* and *ath-5* mutants. **(H)** Slot-blot assays of the overall chloroplast R-loop levels of 21-day-old *atrnh1c*, *1cath-3^{+/+}*, and *1cath-5^{+/+}* plants. RNase H-treated DNA was used as the negative control. R-loops were detected using the RNA:DNA hybrid antibody S9.6 (up panel). DNA loading was stained using SYBR Green (down panel).

4. The authors also pointed that mutation of Pol1A can rescue the growth defects of *atrnh1c* by restricting HO-TRCs in chloroplasts. Given that Pol1A and ATH both play a role in DNA replication and rescue the growth defect of *atrnh1c*, what is the relationship among the ATH, RNH1C and Pol1?

Thanks for your comments. During the DNA replication in the chloroplast genome, the DNA primase ATH synthesizes RNA oligonucleotides used as primers to initiate DNA replication, and the DNA polymerases Pol1A and Pol1B utilize the RNA primers and add nucleotides matched to the template strand in the 5' to 3' direction. Both ATH and Pol1A/B are essential for DNA replication, thus knock-out of ATH or Pol1A/B could lead to embryo lethality in Arabidopsis.

In this study, we found an R166K mutation in ATH can rescue the growth defects in *atrnh1c* by decreasing the RNA primer synthesis and slowing down the replication machinery. Mutation of Pol1A can also lead to similar effects in rescuing the defects in *atrnh1c* by reducing the DNA replication speed. These indicate that reducing the DNA replication speed moderately may mitigate the HO-TRCs in the chloroplast genome, which is supervised by AtRNH1C.

5. Fig 7A, compared with *atrnh1c*, the growth defects of the *1cpol1a* double mutant were partially restored, but *1cpol1b* appears smaller than *atrnh1c*. How to explain this difference of phenotype?

Thank you for pointing out this. Two DNA polymerases, Pol1A and Pol1B, are involved in the replication of the chloroplast genome in Arabidopsis. Previous studies have shown that Pol1B also has functions of DNA repair (Parent et al., 2011, PMID: 21427281), and the *1cpol1b* double mutant displayed more severe chloroplast genome degradation and growth defects than *atrnh1c* (Wang et al., 2021, PMID: 34133716). Compared to Pol1B, Pol1A is a more specific DNA polymerase that is primarily involved in the DNA replication of organelle genomes. Mutation of Pol1A may partially reduce the replication of the chloroplast genome, thus *1cpol1a* double mutant shows a phenotype similar to *acs1*.

Minor points:

1. Fig 1A and 1B, a brief description of *acs1ATH::ATH* and *acs1ATH::ATH-GFP* plants should be included in the text.

Thank you for your valuable advice. We have added a brief description of *acs1^{ATH::ATH}* and *acs1^{ATH::ATH-GFP}* plants in the **Methods (Lines 385-387)**.

2. Fig3C, the size of the marker band is not marked in the figure or labeled in the figure legends.

Thank you for pointing out this omission. We have added the size of the marker band in **Fig. 3C**.

3. Fig 4B and Fig 7E, significant differences were not marked clearly enough.

Thank you for your valuable suggestions. We have modified the corresponding figures to mark the significant differences more clearly. We hope our new manuscript can resolve your concerns.

We would like to thank the reviewers again for taking the time to review our manuscript, and for all the constructive and supportive comments.

REVIEWERS' COMMENTS

Reviewer #1 (Remarks to the Author):

The authors have done well in addressing my comments/suggestions. I propose publication of this study. There is one point I will suggest authors could consider.

The authors can include the response Fig. 8 in the manuscript along with the associated discussion (as in the rebuttal). There has been difference of opinion regarding immunofluorescence efficiency of the antibody used. Clearly authors provide compelling evidence (with data and citations of published literature) supporting their experiments.

Reviewer #2 (Remarks to the Author):

The author has completely solved the questions we raised.

We are grateful to the Reviewers for their valuable comments, which have helped us to improve our manuscript. Detailed point-by-point responses addressing the reviewers' comments and queries are described below in blue.

Reviewer #1's comments and our responses:

The authors have done well in addressing my comments/suggestions. I propose publication of this study. There is one point I will suggest authors could consider.

The authors can include the response Fig. 8 in the manuscript along with the associated discussion (as in the rebuttal). There has been difference of opinion regarding immunofluorescence efficiency of the antibody used. Clearly authors provide compelling evidence (with data and citations of published literature) supporting their experiments.

Thank you very much for your comments. We have followed your suggestions and added the associated discussion about S9.6 immunofluorescence in the manuscript (**Lines 188-196**). Since the response Fig. 8 is the published data, we cannot include it in this article, and we have referenced the corresponding article in our manuscript (Reference 21, PMID: 31914390).

Reviewer #2's comments and our responses:

The author has completely solved the questions we raised.

Thank you very much for your support.

We would like to thank the reviewers again for taking the time to review our manuscript, and for all the constructive and supportive comments.